# Microscopy-based phenotypic profiling of infection by *Staphylococcus aureus* clinical isolates reveals intracellular lifestyle as a prevalent feature

Ines Rodrigues Lopes [1,2], Laura Maria Alcantara[1], Ricardo Jorge Silva [2], Jerome Josse [3], Elena Pedrero Vega[4], Ana Marina Cabrerizo[4], Melanie Bonhomme[3], Daniel Lopez [4], Frederic Laurent[3,5], Francois Vandenesch [3,5], Miguel Mano [2,6,7] ✉ & Ana Eulalio [1,8,9] ✉

*Staphylococcus aureus* is increasingly recognized as a facultative intracellular pathogen, although the significance and pervasiveness of its intracellular lifestyle remain controversial. Here, we applied fluorescence microscopy-based infection assays and automated image analysis to profile the interaction of 191 *S. aureus* isolates from patients with bone/joint infections, bacteremia, and infective endocarditis, with four host cell types, at five times post-infection. This multiparametric analysis revealed that almost all isolates are internalized and that a large fraction replicate and persist within host cells, presenting distinct infection profiles in non-professional vs. professional phagocytes. Phenotypic clustering highlighted interesting sub-groups, including one comprising isolates exhibiting high intracellular replication and inducing delayed host death in vitro and in vivo. These isolates are deficient for the cysteine protease staphopain A. This study establishes *S. aureus* intracellular lifestyle as a prevalent feature of infection, with potential implications for the effective treatment of staphylococcal infections.

*Staphylococcus aureus* (*S. aureus*) is a leading cause of human infections worldwide, causing a broad range of community- and hospital-acquired infections. Disease severity varies widely, ranging from superficial skin infections such as abscesses and impetigo to serious invasive infections, including pneumonia, bacteremia, osteomyelitis, and endocarditis[1,2].

Although *S. aureus* was initially considered an extracellular pathogen, substantial evidence demonstrates that *S. aureus* can adhere, invade and persist within a variety of non-phagocytic host cells, including epithelial and endothelial cells, fibroblasts, osteoblasts, and keratinocytes[3–5]. *S. aureus* adhesion to host cells is mainly dependent on bacterial surface-exposed fibronectin-binding proteins

[1]RNA & Infection Laboratory, Center for Neuroscience and Cell Biology (CNC), University of Coimbra, Coimbra, Portugal. [2]Functional Genomics and RNA-based Therapeutics Laboratory, Center for Neuroscience and Cell Biology (CNC), University of Coimbra, Coimbra, Portugal. [3]Centre International de Recherche en Infectiologie (CIRI), Université de Lyon, Inserm, U1111, Université Claude Bernard Lyon 1, CNRS, UMR5308, ENS de Lyon, Lyon, France. [4]National Centre for Biotechnology, Spanish National Research Council (CNB-CSIC), Madrid, Spain. [5]Centre National de Référence des Staphylocoques, Institut des Agents Infectieux, Hospices Civils de Lyon, Lyon, France. [6]Department of Life Sciences, University of Coimbra, Coimbra, Portugal. [7]British Heart Foundation Centre of Research Excellence, School of Cardiovascular and Metabolic Medicine & Sciences, King's College London, London, United Kingdom. [8]Institute of Biomedicine (iBiMED), Department of Medical Sciences, University of Aveiro, Aveiro, Portugal. [9]Department of Life Sciences, Imperial College London, London, United Kingdom. ✉e-mail: mano@ci.uc.pt; aeulalio@ci.uc.pt

(FnBPs), which engage cell surface fibronectin and integrins, although alternative mechanisms have been identified[6–11]. Invasion of host cells is achieved by a modified zipper-like mechanism, involving extensive F-actin rearrangements. Once internalized, *S. aureus* can have diverse intracellular fates, including rapid clearance, induction of host cell death, intracellular replication, and/or persistence. Of note, bacterial replication and survival have also been reported within professional phagocytes, such as human monocyte-derived macrophages and neutrophils[12,13]. These cells have been suggested as *S. aureus* reservoirs, allowing the persistence of the bacteria in the bloodstream and its metastatic dissemination from the initial inoculation site to other tissues.

Intracellular *S. aureus* is suggested to be a major reservoir for chronic and relapsing staphylococcal infections, being associated with resistance to antibiotic treatment[14]. Along this line, an antibody-antibiotic conjugate able to target intracellular *S. aureus* was shown to be superior to vancomycin in the treatment of *S. aureus* infection[15]. Notwithstanding the increasingly large body of literature documenting *S. aureus* as a facultative intracellular pathogen, the relevance and prevalence of the intracellular lifestyle to the outcome of infection and pathogenesis is yet to be fully elucidated. Work performed to date on intracellular *S. aureus* has been limited to a small number of bacterial strains, using various multiplicities of infection, bacterial growth conditions, and observation times, rendering difficult any global conclusion regarding the pervasiveness of *S. aureus* intracellularity. Indeed, it is still unclear whether the intracellular lifestyle of *S. aureus* is a general feature or a rather infrequent behavior restricted to specific isolates.

Here, we applied a multipronged screening approach based on fluorescence microscopy infection assays and automated image analysis to systematically characterize the infection profile of a collection of 191 *S. aureus* clinical isolates, collected from patients with bone/joint infections, bacteremia, and infective endocarditis. This unbiased analysis was performed at five times post-infection (from invasion up to 48 hpi) and in four host cell types (including professional and non-professional phagocytes). The results obtained reveal that the vast majority of *S. aureus* clinical isolates invade, and that a large fraction of those replicate and persist inside host cells. The prevalence of *S. aureus* intracellularity revealed in this study, and the multiple fates of the bacteria within host cells, offer new perspectives and opportunities for the development of targeted therapeutic strategies to fight staphylococcal infections.

## Results

### Intracellular lifestyle is a predominant feature of infection by *S. aureus* clinical isolates

To systematically and comprehensively analyze the invasion, replication, and persistence of *S. aureus* within host cells, as well as the host cell viability upon infection, we performed fluorescence microscopy-based infection assays in 384-well plates, followed by automated image analysis (Fig. 1a and Supplementary Fig. 1a). Briefly, *S. aureus* clinical isolates were added to host cells for 1 h, extracellular bacteria were eliminated by antibiotic treatment (defined as time-point 0 h), and infection was analyzed at 0.5, 1.5, 3, 6, and 48 h post-infection (hpi), following *S. aureus* labeling with vancomycin BODIPY[16,17]. Automated image analysis extracted multiple features, including the number of *S. aureus* infected cells, cells with high levels of replicating bacteria, and host cell viability, as in our previous studies with other bacterial pathogens[18,19]. We screened a collection of 191 *S. aureus* isolates from patients with bone/joint infections (BJI, 93 isolates), bacteremia without infective endocarditis (Ba, 48 isolates), and infective endocarditis (IE, 50 isolates)[20–22]. The screenings were performed in four distinct host cell types − epithelial cells (HeLa), endothelial cells (EA.hy926), osteoblasts (U2OS), and macrophages (differentiated THP1).

The time course analysis of infection, replication, persistence, and host cell viability for all individual *S. aureus* isolates, as well as the profiles for BJI, Ba, and IE groups, are shown in Fig. 1b–i (full dataset in Supplementary Data 1). Of note, prior to infection *S. aureus* isolates were grown until the exponential growth phase ($OD_{600}$ 0.3-0.8; median 0.50; Supplementary Fig. 1b, c and Supplementary Data 1); we verified that within this range, the parameters analyzed were not influenced by differences in bacterial growth (Supplementary Fig. 1b–d). Three independent experiments were performed for each isolate/host cell/time-point, showing very good reproducibility (Supplementary Fig. 2a, b and Supplementary Data 1).

In both professional phagocytes (macrophages) and non-professional phagocytes (epithelial and endothelial cells, and osteoblasts), the large majority of the *S. aureus* isolates invaded host cells efficiently (Fig. 1b–f and Supplementary Fig. 2c). Indeed, only four isolates presented very low invasion in non-professional phagocytes (<10% of infected cells at 0.5 hpi; 4 of 191, 2.1%); all isolates were internalized in macrophages (minimum 35% infected cells at 0.5 hpi) (Fig. 1b–f and Supplementary Fig. 2c). We hypothesized that the low invasion of these four isolates (Supplementary Fig. 3a) ensued from their inefficient adhesion to host cells, which is mainly dependent on fibronectin-binding proteins (FnBPs) present on the bacteria[9–11]. Using in vitro fibronectin-binding assays, we confirmed that the four isolates were unable to interact with fibronectin (Supplementary Fig. 3b), thus explaining the inefficient invasion exhibited by these isolates. Whole-genome sequencing (WGS) analysis revealed that none of the four isolates harbored the *fnbB* gene. Moreover, 3 of these isolates presented a mutation that leads to a stop codon in *fnbA*. Specifically, isolates BJI006 and BJI010 contain a nucleotide deletion (C, position 1073 downstream of start codon) leading to a premature stop codon (position 1093–1095); isolate IE070 has a 1-nt deletion at position 335 associated with a frameshift that leads to a stop codon (position 349–351). Isolate IE074 presents multiple mutations in both the region described to impact fibrinogen and elastin binding[23] and in the fibronectin-binding domain, although their impact in fibronectin-binding is unclear.

Overall, these results demonstrate that internalization into host cells is a prevalent feature of infection by *S. aureus* clinical isolates, independently of the clinical source of the bacterial isolate and host cell type.

### Replication and persistence within infected cells are recurrent characteristics of *S. aureus* clinical isolates

Although the ability of *S. aureus* to invade host cells is well documented, its capacity to replicate within infected cells has been less characterized. Through the quantification of the integrated intensity and area of the fluorescence signal corresponding to the bacteria at the level of single host cells, we determined the intracellular load of each clinical isolate, for the different time points and host cell types. An appropriate cutoff, validated by complementary approaches (cf. below), was applied to classify cells with a high load of intracellular *S. aureus*, indicative of high intracellular replication (Supplementary Fig. 1a). This analysis revealed that for a significant fraction of the isolates, more than 10% of the host cells presented high levels of *S. aureus* intracellular replication, in at least one of the analyzed times post-infection (Fig. 1b–e, g). This phenotype was particularly evident in macrophages (108 isolates, 56.5%), endothelial cells (93 isolates, 48.7%), and epithelial cells (76 isolates, 40%), and less pronounced in osteoblasts (44 isolates, 23%) (Fig. 1b–e, g and Supplementary Fig. 2c). The comparison of the integrated intensity of the bacterial fluorescence signal at 5 min pi (used as a proxy for bacterial intracellular load at the earliest time-point possible to analyse) with the maximum of intracellular replication per isolate presented low correlation in epithelial cells or macrophages (Supplementary Fig. 2d), indicating that differences in invasion do not explain those observed for replication.

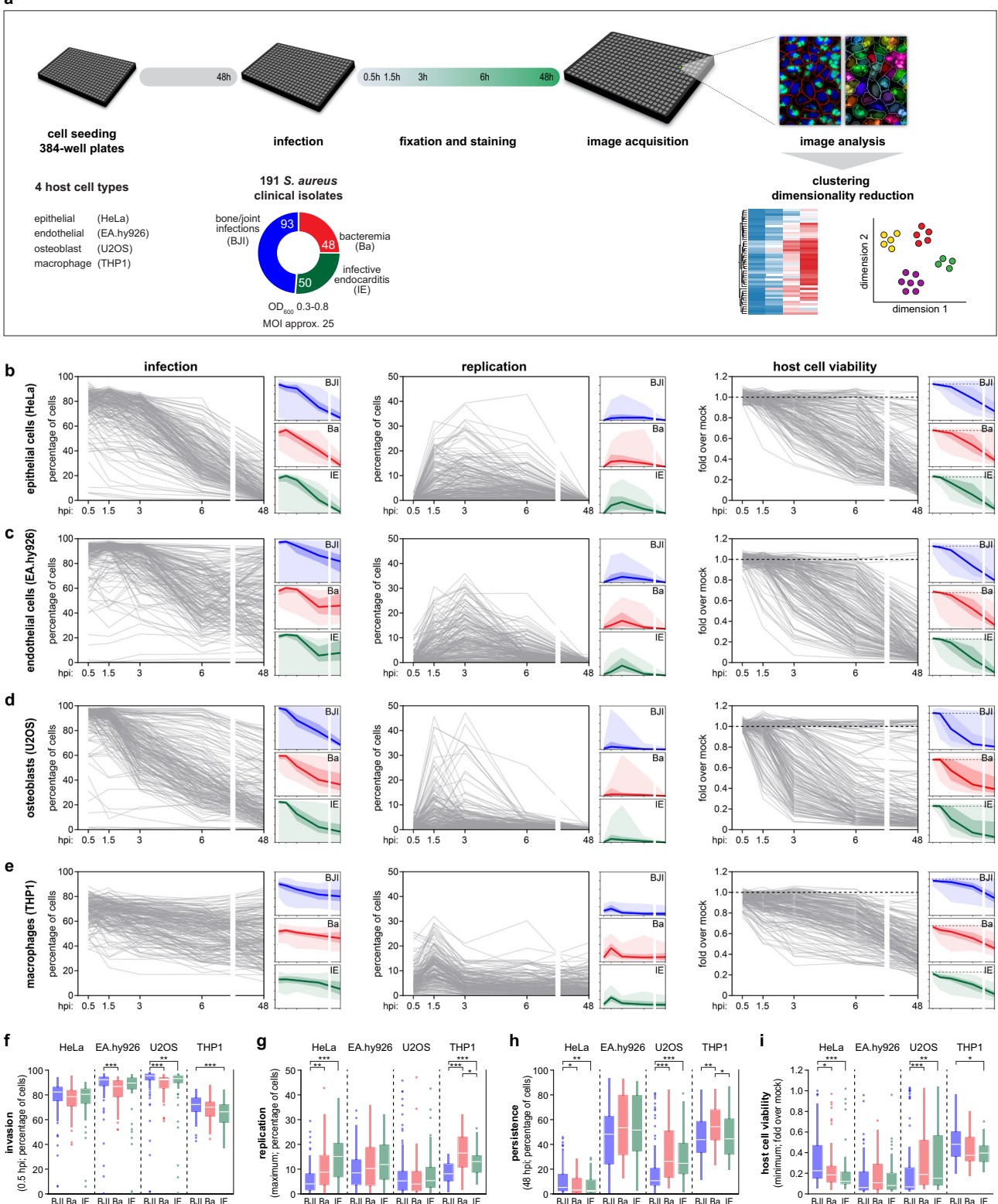

Analysis of the kinetics of *S. aureus* intracellular replication revealed that it occurs typically at relatively early times after infection. For the majority of the isolates, the replication peaked at 1.5 hpi in osteoblasts and macrophages (Fig. 1d, e), and at 3 hpi in endothelial cells (Fig. 1c). In epithelial cells, similar fractions of isolates presented the maximum of replication at 1.5 and 3 hpi (Fig. 1b).

We have also analyzed the extent of infection at 48 hpi (Fig. 1b–e, h), as a proxy for bacterial persistence. Although 48 hpi is an early time-point to assess *S. aureus* persistence, this was the latest time-point compatible with the large-scale microscopy-based infection assays in 384-well format. In epithelial cells, only 58 isolates (30.4%) were present in more than 10% of host cells at 48 hpi (Fig. 1b, h and Supplementary Fig. 2c). This fraction was significantly higher in osteoblasts

**Fig. 1 | Multiparametric analysis reveals that *S. aureus* clinical isolates invade, replicate, and persist in non-professional and professional phagocytic cells.**
**a** Schematic representation of the screening workflow used to evaluate the intracellular lifestyle of 191 *S. aureus* isolates collected from patients with bone/joint infections (BJI, 93 isolates), bacteremia without infective endocarditis (Ba, 48 isolates), and infective endocarditis (IE, 50 isolates). Three independent runs of the screenings for each host cell type were performed. Analysis was performed at 0.5, 1.5, 3, 6, and 48 h post-infection (hpi). **b–e** Time course analysis of infection, intracellular replication, and host cell viability for all individual *S. aureus* isolates in four host cell types, specifically: epithelial cells (HeLa; **b**), endothelial cells (EA.hy926; **c**), osteoblasts (U2OS; **d**), and macrophages (differentiated THP1; **e**); for each isolate, results are presented as the mean of three biologically independent experiments. The subplots show the results for each phenotype stratified by the

clinical source of the *S. aureus* isolates: BJI (blue), Ba (red), and IE (green); lines correspond to the median, 25th–75th percentiles, and minimum-maximum are shown in distinct shades. **f–i** Box-plots showing the data distribution for all *S. aureus* isolates concerning invasion (infection at 0.5 hpi; **f**), intracellular replication (maximum value; **g**), persistence (infection at 48 hpi; **h**), and host cell viability (minimum value; **i**). Results are shown for the four cell types tested and stratified by the clinical source of the *S. aureus* isolates; box-plots were generated using the mean of three biologically independent experiments per clinical isolate; white lines show the medians, box limits indicate the 25th–75th percentiles, whiskers extend 1.5 times the interquartile range from the 25th and 75th percentiles. *$P < 0.05$, **$P < 0.01$, and ***$P < 0.001$ (statistical analysis is detailed in Supplementary Data 3).

(138 isolates, 72.3%), endothelial cells (184 isolates, 96.3%), and macrophages (191 isolates, 100%) (Fig. 1c–e, h and Supplementary Fig. 2c).

Our analysis also shows that infection has a strong impact on host cell viability, particularly in non-professional phagocytes (Fig. 1b–e, i). In these three cell types, cell viability was reduced to less than 20% of mock-treated cells upon infection with the majority of the isolates, specifically 88 isolates (53.9%) in epithelial cells, 121 isolates (63.4%) in osteoblasts, and 141 isolates (73.8%) in endothelial cells (Fig. 1b–e, i). In macrophages, a milder reduction of cell viability was observed, with only three isolates reducing viability to less than 20% of the mock-treated cells. In terms of kinetics, a drop in host cell viability was typically observed following the peak of replication (Fig. 1b–e).

The comparison of the three subsets of *S. aureus* isolates based on clinical source (BJI, Ba, and IE) revealed clear differences (Fig. 1b–i). In macrophages and epithelial cells, the Ba and IE isolates presented higher replication than the BJI isolates, while in osteoblasts, there were no obvious differences (Fig. 1b–e, g). In addition, in osteoblasts, the BJI isolates presented lower persistence than Ba and IE isolates, and had a stronger effect on reducing host cell viability (Fig. 1b–e, 1h, i). Ba and IE isolates presented comparable phenotypes in all cell types, except for macrophages in which Ba isolates presented higher replication and persistence than IE isolates (Fig. 1b–i).

Together, these results highlight intracellular replication and persistence as recurring characteristics of infection by *S. aureus* isolates, showing distinct extent and kinetics depending on the host cell type and *S. aureus* clinical source.

## *S. aureus* clinical isolates present distinctive intracellular lifestyles upon infection of professional and non-professional phagocytes

As described above, the global kinetics and extent of replication, persistence, and host cell viability upon infection with *S. aureus* clinical isolates are highly dependent on the host cells. Nonetheless, the comparison of the phenotypic profiles of individual isolates enables the identification of commonalities and dissimilarities depending on the host cell type.

Hierarchical cluster analysis revealed several groups of isolates with interesting infection profiles (Fig. 2a). Two clusters of isolates showed very low or moderate invasion of non-professional phagocytes (clusters I and II, respectively). Cluster I contains the four isolates described above showing very low invasion (<10% of infection in HeLa, EA.hy926, and U2OS at 0.5 hpi; Fig. 1b–d and Supplementary Fig. 3a) and an additional isolate (isolate BJI026) that shows low invasion in epithelial and endothelial cells; isolates from cluster I did not induce host cell toxicity. Isolates in cluster II presented moderate invasion and/or rapid clearance, very low intracellular replication, and induced low host cell death in non-professional phagocytes. All isolates belonging to the remaining clusters efficiently invaded non-professional phagocytes (clusters III, IV, and V, with cluster IV divided into four subclusters), exhibiting various levels of intracellular replication and persistence at 48 hpi, and distinct kinetics and levels of

host cell death (Fig. 2a). Particularly interesting are: (i) isolates from cluster III, which showed high persistence at 48 hpi, low intracellular replication, and low host cytotoxicity in non-professional phagocytes; and isolates belonging to cluster V, which displayed very high replication but presented delayed host cell death in non-professional phagocytes when compared to isolates with comparable high levels of replication (cluster IVb). A schematic representation summarizing the observed intracellular phenotypes is shown in Fig. 2b, and representative microscopy images of isolates belonging to the various clusters are shown in Fig. 3a–d and Supplementary Fig. 4a–d.

Notably, the clusters of isolates described above are mostly defined by their intracellular phenotypes in non-professional phagocytes, while presenting a higher heterogeneity of phenotypes in macrophages. Indeed, pairwise comparison of the maximum intracellular replication or intracellular persistence at 48 hpi per isolate in the various host cell types revealed a high correlation between osteoblasts, epithelial and endothelial cells (Spearman's $r > 0.60$; Fig. 2c, d), while considerably lower correlations were observed for macrophages (Spearman's $r < 0.42$; Fig. 2c, d). Similarly, pairwise comparison of the invasion of *S. aureus* isolates in the different cell lines showed good correlations between osteoblasts, epithelial and endothelial cells (Spearman's $r > 0.58$; Supplementary Fig. 4e), which were significantly lower when compared with macrophages (Spearman's $r < 0.46$; Supplementary Fig. 4e). This likely reflects the different internalization mechanisms used by non-professional phagocytic cells (mediated by bacterial and host proteins) and macrophages (phagocytosis). Concerning host cell viability, pairwise analysis of the minimum of cell viability observed upon infection with each isolate in the different cell types showed moderate correlations (Spearman's $r < 0.50$; Supplementary Fig. 4f), except for the comparisons between endothelial cells and osteoblasts (Spearman's $r$ 0.812) and epithelial and endothelial cells (Spearman's $r$ 0.588). Consistent with the low phenotypic correlation between infection of macrophages and non-professional phagocytic cells, clustering performed based solely on the phenotypes observed in macrophages yielded groups of isolates clearly distinct from those obtained when considering the phenotypes in the four cell types (Fig. 2a and Supplementary Fig. 5a, b).

Collectively, these results demonstrate that the intracellular lifestyles of individual *S. aureus* isolates in non-phagocytic cells are, in broad lines, comparable, but distinct from those in macrophages.

## *S. aureus* isolates group based on distinctive intracellular phenotypes

Given the diversity of the phenotypes exhibited by the 191 *S. aureus* isolates, we applied t-distributed stochastic neighbor embedding (t-SNE)[24] to better visualize distinct phenotypic classes (Fig. 4a).

The first observation from the t-SNE representation was the separation between BJI vs. Ba and IE isolates (Fig. 4a). The segregation of isolates from distinct clinical sources based on phenotypic differences is particularly interesting, given that whole-genome phylogenetic analysis was unable to achieve this type of discrimination

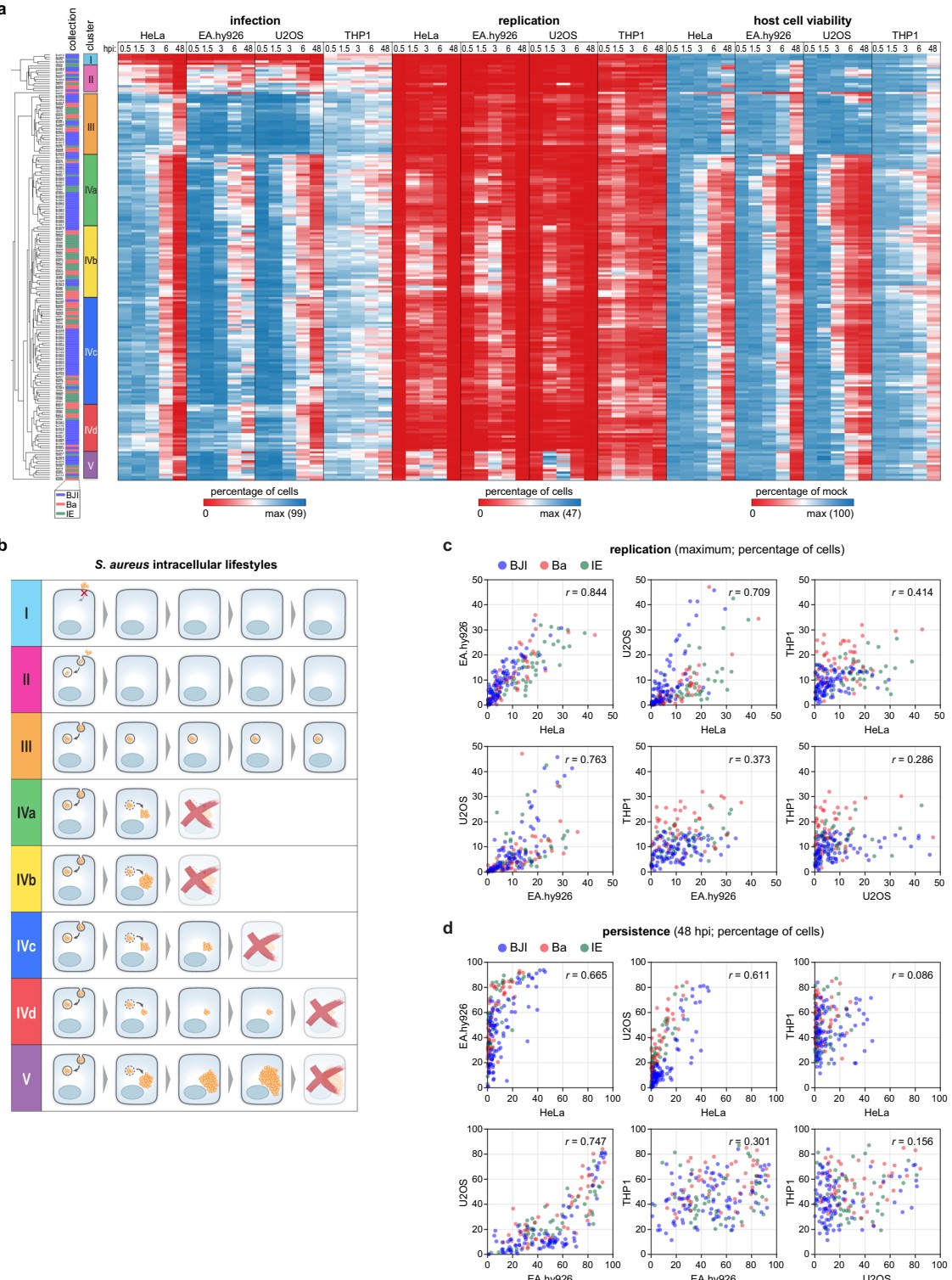

**Fig. 2 | Individual *S. aureus* isolates exhibit similar intracellular lifestyles in non-phagocytic cells, but distinct from those in macrophages. a** Heat map showing the phenotypic profiles (infection, intracellular replication, and host cell viability) of the 191 *S. aureus* isolates following infection of non-phagocytic (HeLa, EA.hy926, and U2OS) and phagocytic (differentiated THP1) cells, at five times post-infection (0.5, 1.5, 3, 6, and 48 hpi). Results are presented as the mean of three biologically independent experiments. Hierarchical clustering of the phenotypic profiles exhibited by the bacterial isolates was performed based on Euclidean distance. Eight distinct infection profiles were identified (five clusters, with cluster IV divided into four subclusters). **b** Schematic representation of the main *S. aureus*

phenotypic features (infection, replication, persistence, host cell viability) associated with each cluster. **c, d** Pairwise comparison of the percentage of cells with high *S. aureus* intracellular replication (maximum value; **c**) and persistence (infection at 48 hpi; **d**) upon infection of epithelial cells (HeLa), endothelial cells (EA.hy926), osteoblasts (U2OS), and macrophages (THP1). The colors of datapoints correspond to the clinical origin of the *S. aureus* isolates. Results are presented as the mean of three biologically independent experiments. Spearman's rank correlation coefficients are shown in the upper right corner of each graph.

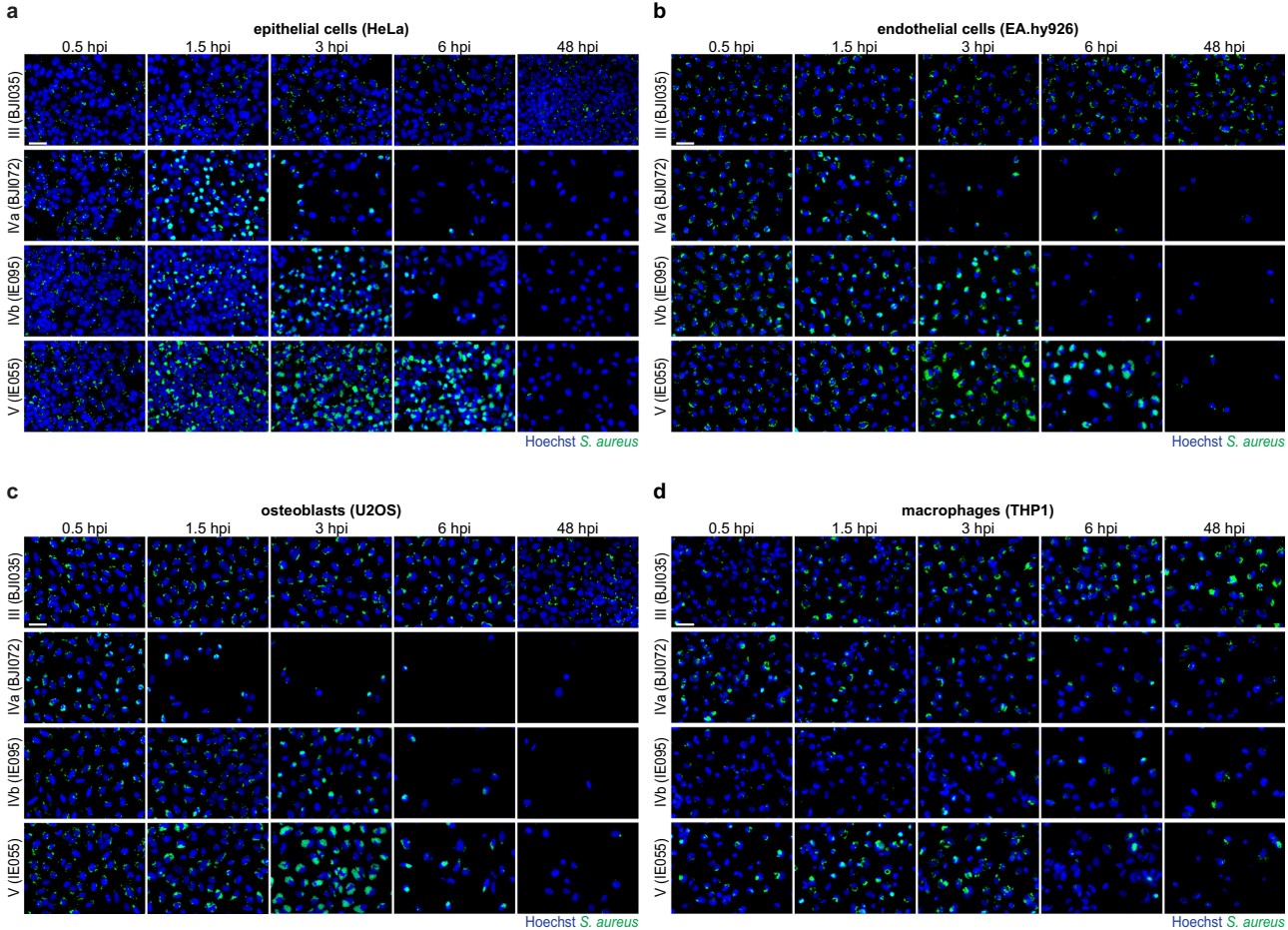

**Fig. 3 | Intracellular lifestyles of *S. aureus* isolates belonging to distinct phenotypic clusters. a–d** Representative fluorescence microscopy images of infection of epithelial cells (HeLa; **a**), endothelial cells (EA.hy926; **b**), osteoblasts (U2OS; **c**), and macrophages (THP1; **d**) with selected *S. aureus* isolates belonging to four phenotypic profile clusters identified in Fig. 2a (clusters III, IVa, IVb, and V), at five times post-infection (0.5, 1.5, 3, 6, and 48 hpi). Images are representative of three biologically independent experiments. Scale bar, 50 µm. Fluorescence microscopy images for infection with S. *aureus* isolates belonging to the remaining clusters are shown in Supplementary Figs. 3a, 4a–d.

(Fig. 4b). Based on WGS of the 191 isolates, 37 sequence types (STs) were identified and assigned to 23 distinct clonal complexes (CCs); none of the STs or CCs was associated with a particular clinical source (Supplementary Data 1).

The groups of isolates identified using the hierarchical clustering were clearly segregated in the t-SNE visualization (Supplementary Fig. 6a). For example, the non-invasive isolates (cluster I, highlighted in cyan in Fig. 4a and Supplementary Fig. 6a) grouped at one end of the t-SNE, neighbored by cluster II isolates that present moderate invasion (pink in Supplementary Fig. 6a). Also noteworthy was the observation of two well-separated groups of *S. aureus* isolates (highlighted in orange and purple in Fig. 4a and Supplementary Fig. 6a), which corresponded to isolates from clusters III and V (Fig. 2a). Notably, the relative proportion of BJI, Ba, and IE isolates was maintained in groups III and V, indicating that these phenotypes are independent of the clinical origin. The isolates Ba009 and Ba037 that were not included in a cluster based on hierarchical clustering (Fig. 2a) were assigned to clusters V and IVa, respectively, based on the t-SNE (white in Supplementary Fig. 6a). Of note, no correlation was observed between phenotypic clustering and genome-based phylogenetic analysis (Supplementary Figs. 6b, c, 5c).

The main phenotypic features of the distinct groups are shown in Fig. 4c–f and Supplementary Fig. 6d–h. The results of the microscopy-based infection assays were validated in epithelial cells using a subset of bacterial isolates belonging to each phenotypic cluster (four isolates from cluster I and five isolates from each of the other clusters, total 39 isolates) using alternative experimental approaches, specifically: (i) colony forming unit (CFU) assays for intracellular bacterial load (Supplementary Fig. 7a); and (ii) lumino-genic ATP assay to assess host cell viability (Supplementary Fig. 7b). The eFluor proliferation assay[25,26], based on dilution of a fluorescence dye with bacterial replication, was also applied to a few *S. aureus* isolates (BJI035 – cluster III, IE095 – cluster IVb, and BJI008 – cluster V) to further evaluate bacterial replication within infected cells (Supplementary Fig. 7c). We observed that at 0.5 hpi all the bacteria were eFluor-670 positive (co-localizing with vancomycin BODIPY); at 3 hpi, eFluor-670 negative bacteria were observed for isolates from clusters IVb and V (IE095 and BJI008, indicating bacterial replication), whereas the isolate from cluster III remained eFluor-670 positive (BJI035, indicating lack of replication). We also performed time-lapse microscopy with a small subset of *S. aureus* strains/isolates expressing GFP (USA300, BJI035 – cluster III, BJI009 – cluster IVa, BJI008 – cluster V; Supplementary Videos 1–4). Taken together, these data fully corroborate the main readouts and findings obtained using microscopy-based infection assays.

Considering the inherent limitations of using transformed cell lines, we have analysed infection in primary cells, specifically human umbilical vein endothelial cells (HUVEC), using the subset of 39 isolates (Supplementary Fig. 7d–i). Globally, comparable infection phenotypes were obtained for isolates belonging to each phenotypic

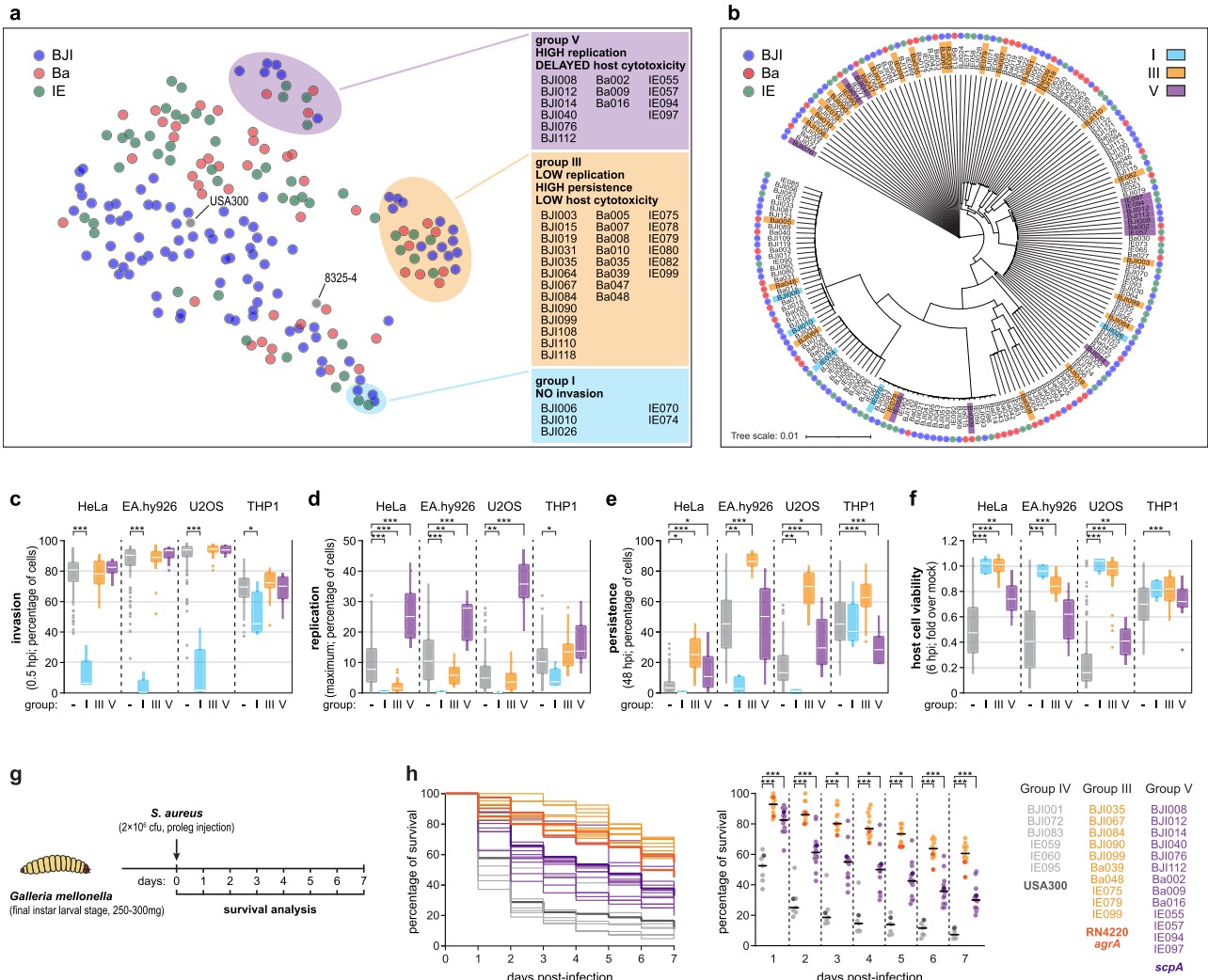

**Fig. 4 | *S. aureus* isolates from distinct clinical sources segregate based on phenotypic profiles. a** t-SNE visualization of the 191 *S. aureus* isolates based on individual multiparametric phenotypic profiles comprising data of infection, intracellular replication, and host cell viability, in four cell types, at five times post-infection (total 60 features). Each circle represents a single *S. aureus* isolate; circles are colored according to the clinical origin of the isolate; reference strains (USA300 and 8325-4) are colored in gray. Bacterial isolates belonging to three groups are highlighted in cyan, orange, and purple (groups I, III, and V, respectively), as defined by hierarchical cluster analysis (Fig. 2a). The identity of the isolates belonging to each of these groups is indicated in the colored boxes. t-SNE visualization of all bacterial isolates according to the phenotypic cluster is shown in Supplementary Fig. 6a. **b** Phylogenetic tree of the 191 *S. aureus* clinical isolates based on whole-genome sequencing (core genome). The color of the circles on the outermost ring indicates the clinical origin of the *S. aureus* isolates. Isolates from groups I, III, and V are highlighted. **c–f** Box-plots showing the distribution of data for *S. aureus* isolates concerning invasion (infection at 0.5 hpi; **c**), intracellular replication (maximum value; **d**), persistence (infection at 48 hpi; **e**), and host cell viability (6 hpi; **f**). Results are shown for the four cell types tested and stratified by phenotypic group, specifically groups I (cyan), III (orange), V (purple), and remaining isolates belonging to groups II and IV (gray); box-plots were generated using the mean of three biologically independent experiments per clinical isolate; white lines show the medians, box limits indicate the 25th–75th percentiles, whiskers extend 1.5 times the interquartile range from the 25th and 75th percentiles. Results for all the individual groups are shown in Supplementary Fig. 6d–h. *$P < 0.05$, **$P < 0.01$, and ***$P < 0.001$ (statistical analysis is detailed in Supplementary Data 3). **g** Schematic representation of the workflow used for the *Galleria mellonella* larvae infection experiments. **h** Seven-day survival curves and dot-plot summarizing larvae survival after infection with *S. aureus* isolates selected from clusters IVa, IVb, IVc (six isolates; gray), isolates belonging to groups III (ten isolates; orange), and group V (13 isolates; purple). *S. aureus* USA300 and strains lacking *agr* activity (RN4220 and *agrA* mutant) or staphopain A (*scpA* mutant) were used as controls and are shown in darker lines/dots. Results are presented as the mean of four biologically independent experiments performed with ten larvae per clinical isolate/strain; black lines in the dot-plot correspond to the medians. *$P < 0.05$ and ***$P < 0.001$ (statistical analysis is detailed in Supplementary Data 3). Source data are provided as a Source Data file.

cluster in HUVEC and in the non-professional phagocytic cells used for the large-scale screenings.

Given the strength of the phenotypes of the isolates belonging to groups III and V in vitro (Fig. 4c–f), we sought to evaluate if this could also be observed in vivo. For this, we used the *Galleria mellonella* larvae infection model (Fig. 4g), which has been widely used to study microbial pathogenesis/virulence, including that of *S. aureus*[27–30]. Comparative analysis was performed between isolates belonging to groups III (10 out of 27 isolates) and V (13 out of 13

isolates), and a subset of isolates that included *S. aureus* USA300 and six isolates presenting phenotypes similar to USA300 in vitro, i.e., high replication, and high host cell death (selected from groups IVa, IVb, IVc; Fig. 2a). USA300 and isolates with a similar in vitro phenotypic profile elicited a marked reduction of larvae survival (gray in Fig. 4h). On the contrary, the isolates belonging to groups III (orange in Fig. 4h) and V (purple in Fig. 4h) showed significantly lower virulence. These survival results are in complete agreement with the in vitro observations of lower or delayed host cell death induced

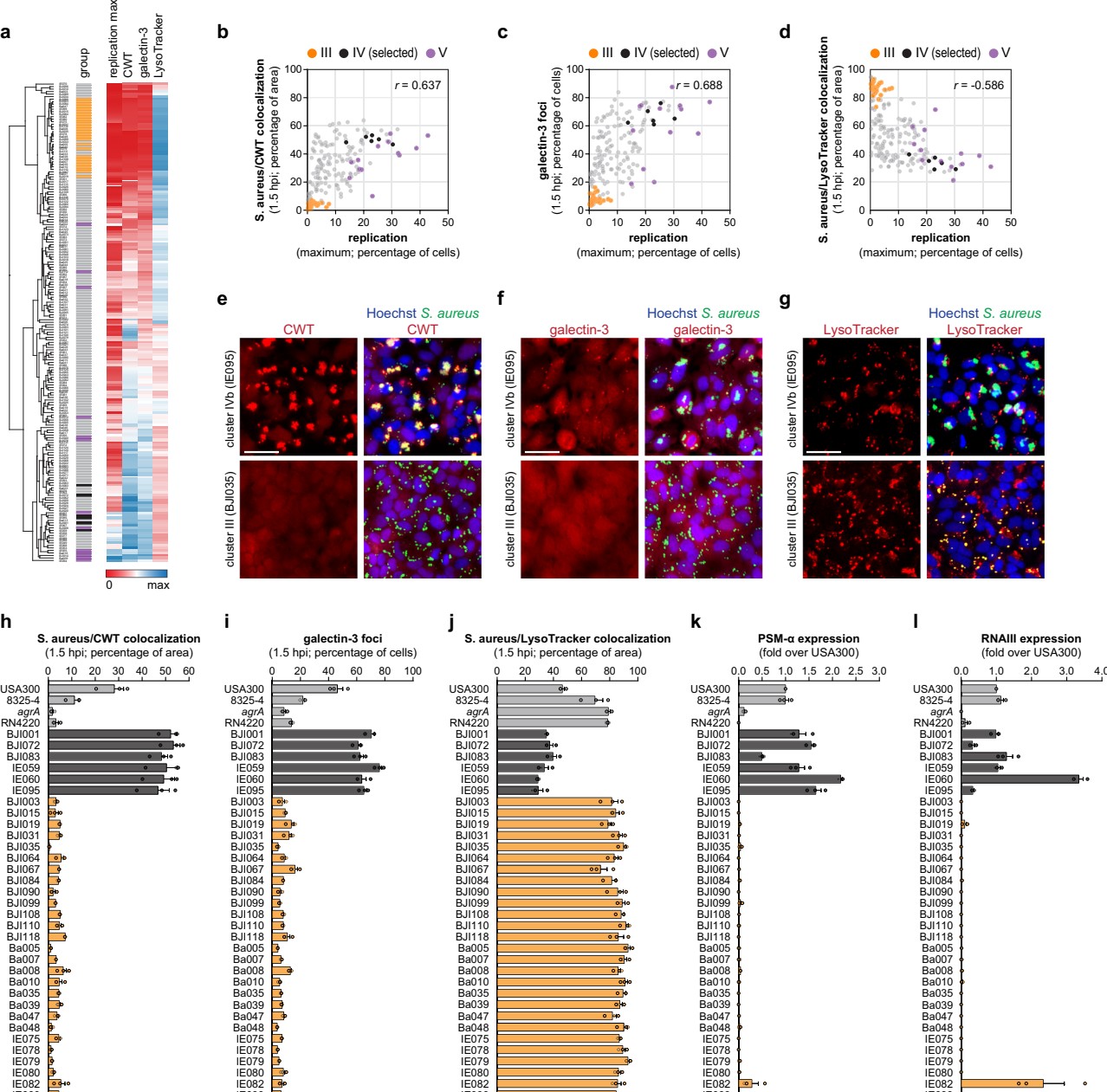

**Fig. 5 | *S. aureus* isolates presenting low vacuolar escape and intracellular replication upon infection of non-phagocytic host cells are *agr* system defective. a** Heat map showing the percentage of cells with high *S. aureus* intracellular replication (maximum value), percentage of *S. aureus*/CWT colocalization (at 1.5 hpi), percentage of infected cells positive for galectin-3 foci (at 1.5 hpi), and percentage of *S. aureus*/LysoTracker colocalization (at 1.5 hpi), for epithelial cells (HeLa) infected with the 191 *S. aureus* isolates. The results shown are the mean of three biologically independent experiments. Hierarchical clustering was performed based on Euclidean distance. Group III and V bacterial isolates are highlighted in orange and purple, respectively; six selected isolates showing high replication and high host cell death (selected from clusters IVa, IVb, IVc) are highlighted in black; all the other isolates are shown in light gray. **b–d** Pairwise comparison of *S. aureus* intracellular replication with *S. aureus*/CWT colocalization (**b**), or infected cells positive for galectin-3 foci (**c**), or *S. aureus*/lysotracker colocalization (**d**). The 191 S. aureus isolates are color-coded as described in panel **a**. Spearman's rank correlation coefficients are shown in the upper right corner of each graph.

**e–g** Representative fluorescence microscopy images of HeLa cells infected with *S. aureus* isolates belonging to cluster III (BJI035) and IVb (IE095), to evaluate the recruitment of fluorescently-labeled CWT (**e**), formation of galectin-3 foci (**f**), and colocalization with LysoTracker (**g**). Microscopy images are representative of three biologically independent experiments and correspond to 1.5 hpi. Scale bar, 50 μm. **h–j** Quantification of the percentages of *S. aureus*/CWT colocalization (**h**), infected cells positive for galectin-3 foci (**i**), *S. aureus*/LysoTracker colocalization (**j**), for HeLa cells infected with *S. aureus* isolates belonging to cluster III (orange), six isolates selected from clusters IVa, IVb, IVc (black), and four control *S. aureus* strains (USA300, 8325-4, *agrA* mutant, RN4220; light gray). All results correspond to 1.5 hpi and are shown as mean ± s.e.m of three biologically independent experiments. **k, l.** Expression levels of PSM-α (**k**) and RNAIII (**l**) determined by qRT-PCR in bacterial cultures corresponding to the *S. aureus* isolates described in panels **h–j** Results are shown normalized to *S. aureus* USA300, and presented as mean ± s.e.m of three biologically independent experiments. Source data are provided as a Source Data file.

upon infection with the *S. aureus* isolates belonging to groups III and V, respectively.

Overall, the phenotypic profiling highlighted interesting groups of *S. aureus* isolates exhibiting distinctive intracellular lifestyles upon infection of non-professional phagocytes, which likely reflect specific genetic traits.

### Low vacuolar escape and replication within non-phagocytic host cells is associated with *agr* system deficiency

Various intracellular bacterial pathogens escape the vacuole to avoid the harsh environment of the phagolysosome and reach the nutrient-rich cytosol[31,32]. *S. aureus* was shown to perform vacuolar escape in non-professional phagocytic cells[33–35], although this was suggested to be isolate-dependent[36,37].

To evaluate the vacuolar vs. cytoplasmic localization of the 191 *S. aureus* clinical isolates in epithelial cells (HeLa cells), we applied three complementary approaches: (i) evaluation of the recruitment of fluorescently-labeled C-terminal cell wall-targeting domain (CWT) of lysostaphin to *S. aureus* cell wall, indicative of phagosomal membrane rupture[33,35,38]; (ii) quantification of infected cells with galectin-3 foci, used as a marker for damaged endo-lysosomes and bacterial-induced ruptured vacuoles[39–41]; and (iii) colocalization of *S. aureus* with Lyso-Tracker, a marker of endo-lysosomes (Fig. 5a and Supplementary Data 2). For the two first assays, HeLa cells stably expressing mRFP-CWT or mCherry-galectin-3 were used. As expected, there was a strong correlation between the percentage of *S. aureus*/CWT colocalization and the percentage of infected cells positive for galectin-3 foci (Spearman's $r$ 0.905; Supplementary Fig. 8a). Both parameters inversely correlated with the percentage of *S. aureus*/LysoTracker colocalization (Spearman's $r$ −0.855 and −0.861, respectively; Supplementary Fig. 8b, c). Comparison of these datasets with the maximal replication of each isolate showed that isolates presenting high intracellular replication display a high degree of colocalization with CWT (Fig. 5a, b, e) and a high percentage of cells positive for galectin-3 foci (Fig. 5a, c, f), while showing low colocalization with LysoTracker (Fig. 5a, d, g), altogether demonstrating the cytosolic localization of these isolates. In contrast, isolates with low intracellular replication were predominantly vacuolar, as indicated by the low CWT colocalization/recruitment (Fig. 5a, b, e), low positivity for galectin-3 foci (Fig. 5a, c, f), and high colocalization with LysoTracker (Fig. 5a, d, g). In addition, an inverse correlation was observed between *S. aureus*/CWT colocalization and host cell viability (Supplementary Fig. 8d, e), corroborating the proposed association between vacuolar escape and induction of host cell death[34,36,42].

Multiple studies have ascribed the phagosomal escape to the *S. aureus* global virulence regulator *agr*[5,32]. In particular, the α-type phenol-soluble modulins (α-type PSMs), which are transcriptionally controlled by the *agr* system[43], were identified as key players of vacuolar escape[33]. Considering that the isolates belonging to group III were among those exhibiting more vacuolar localization (i.e., low CWT colocalization, low galectin-3, high LysoTracker; Fig. 5a–g), we hypothesize that these isolates could be deficient in the *agr* system. In agreement, the results for group III isolates were similar to strains lacking *agr* activity, namely RN4220 and *agrA* transposon insertion mutant (Fig. 5h–j and Supplementary Fig. 8f–h); both strains mapped to group III in the t-SNE visualization (Supplementary Fig. 8i). Accordingly, the RN4220 and *agrA* mutant strains showed reduced virulence in the *Galleria* model (Fig. 4h). Analysis of the expression of *agr* effectors or downstream genes (RNAIII and PSM-α, respectively) demonstrated that the *S. aureus* group III isolates are indeed deficient in the *agr* system (Fig. 5k, l).

Together, these results show that the *S. aureus* isolates that efficiently invade and persist within non-phagocytic host cells, but do not reach the host cytosol or replicate, are defective in the *agr* system.

### Absence of staphopain A enables *S. aureus* sustained intracellular replication and delayed onset of host cell death

Various studies have indicated a connection between the translocation of *S. aureus* to the cytoplasm, followed by cytoplasmic replication and the induction of death in non-professional phagocytic host cells[34,36,42]. Globally, our results support this trend by showing an inverse correlation between replication and host cell viability (Fig. 6a, b and Supplementary Fig. 9a), particularly in epithelial and endothelial cells. Intriguingly, our multiparametric analysis highlighted that *S. aureus* isolates from group V present high and sustained replication within host cells but induce delayed host cell death in the three non-professional phagocytic cells (highlighted in purple in Fig. 6a, b and Supplementary Fig. 9a) when compared with USA300 and other isolates with high vacuolar escape and replication (highlighted in black in Fig. 6a, b and Supplementary Fig. 9a). This is particularly noticeable at 3 and 6 hpi.

Of note, isolates from group V were able to escape the vacuole at efficiencies comparable to USA300 and other isolates showing high replication (Fig. 5a–d), suggesting that their ability to replicate at high levels while inducing delayed host cell toxicity is dependent on another specific trait. Recently, staphopain A, one of the two cysteine proteases annotated in the *S. aureus* genome, was implicated in host cell death upon *S. aureus* infection[44]. Interestingly, analysis of the gene encoding staphopain A (*scpA*) in the isolates belonging to group V revealed that: (i) 7 out of 13 isolates (BJI008, BJI012, BJI112, Ba002, IE057, IE094, and IE097) present a mutation (G-T) located 57 nucleotides upstream of the start codon, in the predicted −35 element of the promoter; (ii) three isolates (BJI014, Ba009, and IE055) bear point mutations in the Shine-Dalgarno sequence in position −10/−11 upstream of start codon; (iii) isolate BJI076 has a mutation in the *scpA* start codon; (iv) isolate BJI040 presents a 1-nt deletion at a stretch of A (position 532–536) in the *scpA* ORF, resulting in a premature stop codon. No relevant mutations in the *scpA* gene were found for isolate Ba016. The gene *scpA* is part of the operon *scpAB* that encodes staphopain A and its endogenous inhibitor staphostatin A (*scpB*). Staphopain A is secreted as a zymogen and activated by autolytic cleavage[45]. *S. aureus* protects itself from proteolytic degradation by producing staphostatin A, which prevents premature autocatalytic activation by stabilizing the proStaphopain[46]. In addition to its role in cell death, staphopain A was shown to cleave several human proteins in vitro, including fibrinogen and collagen, to modulate biofilm integrity and to mediate evasion of the immune system[47–50].

Consistent with the absence of staphopain A being responsible for the delayed host cell death observed for group V isolates, the mature form of the protease was not detected in the supernatants of liquid cultures of these isolates (13 out of 13), similar to *scpA S. aureus* transposon insertion mutant (Fig. 6c). This is in contrast with USA300 and isolates presenting high replication and high host cell death (e.g., BJI001, BJI009, BJI072, Ba046, IE053, IE059, IE060, IE093, and IE095; Fig. 6c and Supplementary Fig. 9b). Of note, the infection profiles of the *scpA* mutant strain both in vitro (Fig. 6d–f and Supplementary Figs. 8i, 10a–f) and in vivo (Fig. 4h) were similar to those of group V isolates.

Notably, the high replication and delayed host cell death phenotypes exhibited by isolates from group V, as well as by the *scpA* mutant strain, were counteracted by the ectopic expression of staphopain A from a plasmid expressing the *scpAB* operon under the control of its endogenous promoter (pScpAB; Fig. 6d–f and Supplementary Fig. 10a–g). The kinetics of replication and host cell death was, in some instances, faster in the isolates ectopically expressing staphopain A than in wild-type *S. aureus* isolates natively expressing staphopain A (Figs. 3, 6d and Supplementary Fig. 10a, b), likely due to a higher expression of the protease driven by the high-copy plasmid (Supplementary Fig. 10g). Corroborating the relevance of staphopain A to the phenotypes observed, treatment of host cells with the cell-permeable

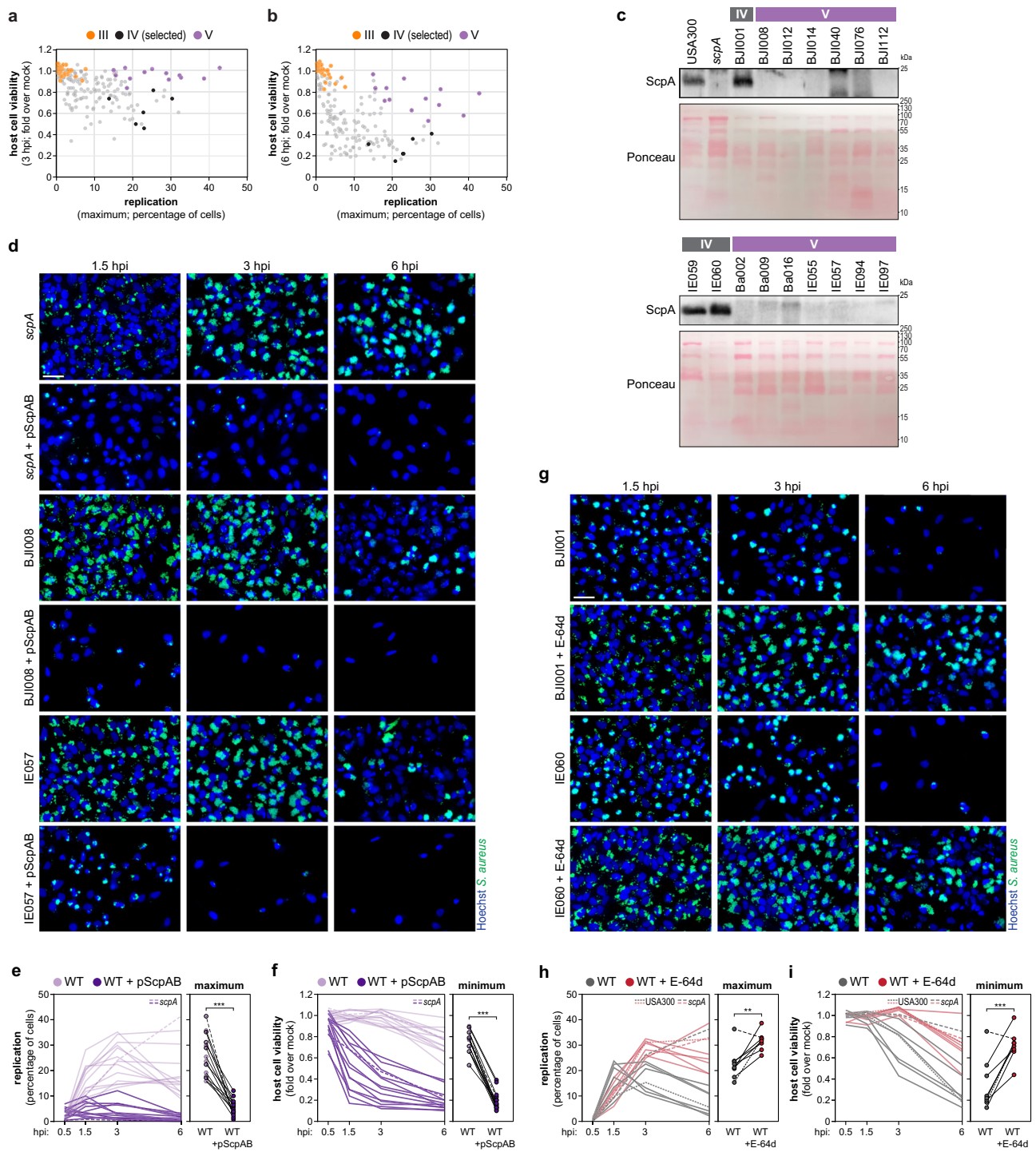

inhibitor of cysteine proteases E-64d resulted in a significant reduction of host cell death and increased intracellular replication of isolates that natively express staphopain A (USA300 and six selected isolates belonging to clusters IVa, IVb, IVc; Fig. 6g–i and Supplementary Fig. 10h). Conversely, the *scpA* mutant strain was not affected by E-64d treatment (Supplementary Fig. 10h). Even though E-64d can also target host cysteine proteases and inhibition of host cysteine proteases has been shown to affect the course of bacterial infection, e.g., *Salmonella*[51], taken together our results strongly indicate that for *S. aureus* the described phenotypes are related to inhibition of staphopain A activity and not of host cysteine proteases.

Overall, these results show that the *S. aureus* group V isolates are deficient in staphopain A expression and that the loss of activity of this

protease results in prolonged high levels of *S. aureus* intracellular replication and delayed onset of host cell death.

## Discussion

*S. aureus* has historically been considered an extracellular pathogen. However, accumulating evidence in vitro and in vivo demonstrates that *S. aureus* can invade, survive and replicate in a wide range of host cell types[3–5,12,13]. Importantly, until now, these studies were limited to a small number of *S. aureus* strains, focused on infection of single host cell types, and/or frequently performed using different infection protocols and endpoints. As such, the pervasiveness and relevance of the *S. aureus* intracellularity remained contentious. In this study, we report a systematic, multiparametric analysis of *S. aureus* intracellular lifestyle

**Fig. 6 | *S. aureus* isolates showing high intracellular replication and delayed host cell death are deficient for the cysteine protease staphopain A.**
**a**, **b** Comparison between the percentage of cells with high *S. aureus* intracellular replication (maximum) and the viability of epithelial cells (HeLa) infected with the 191 *S. aureus* isolates, at 3 (**a**) and 6 hpi (**b**). Results are shown as mean of three biologically independent experiments. Group III and V bacterial isolates are highlighted in orange and purple, respectively; six isolates selected from clusters IVa, IVb, and IVc are highlighted in black; all the other isolates are shown in light gray.
**c** Staphopain A protein levels, determined by western blot, in the supernatant of bacterial liquid cultures of the 13 *S. aureus* isolates belonging to group V. *S. aureus* USA300; *scpA* mutant, and isolates BJI001, IE059, IE060 (belonging to clusters IVa and IVb) were used as controls. Western blots are representative of three independent experiments. Ponceau staining of the membranes is shown.
**d–f** Representative fluorescence microscopy images (**d**), and time course quantification of the percentage of host cells with high bacterial intracellular replication (**e**), and host cell viability (**f**), of HeLa cells infected with *S. aureus* isolates belonging to cluster V (WT, light purple in panels **e**, **f**) or with isolates modified to ectopically express both staphopain A and staphostatin A from a plasmid (WT + pScpAB, dark

purple in panels **e**, **f**). *S. aureus scpA* mutant and *scpA* + pScpAB are shown for comparison (dotted lines). Results shown in panels **e** and **f** are the mean of three biologically independent experiments; rightmost plots indicate the maximum of replication (**e**) or minimum of host cell viability (**f**) for WT or modified bacteria; ***$P < 0.001$ (statistical analysis is detailed in Supplementary Data 3). Scale bar, 50 μm. **g–i** Representative fluorescence microscopy images (**g**), and time course quantification of the percentage of host cells with high bacterial intracellular replication (**h**) and host cell viability (**i**), of HeLa cells infected with *S. aureus* isolates that natively express staphopain A (selected from clusters IVa, IVb, IVc). Infection was performed without (dark gray) or with treatment with the inhibitor of cysteine proteases E-64d (dark red). *S. aureus* USA300 and *scpA* mutant are shown for comparison in panels **h**, **i** (dotted lines). Results shown in panels **h** and **i** are the mean of three biologically independent experiments; rightmost plots indicate the maximum of replication (**h**) or a minimum of host cell viability (**i**) in the presence/absence of the inhibitor; **$P < 0.01$ and ***$P < 0.001$ (statistical analysis is detailed in Supplementary Data 3). Scale bar, 50 μm. Source data are provided as a Source Data file.

(infection, replication, persistence, and host cell death) by the application of a series of microscopy-based screenings of 191 *S. aureus* isolates from three clinical sources (bone/joint infections, bacteremia, and infective endocarditis), in four host cell types (epithelial, endothelial, osteoblasts, and macrophages, i.e., non-professional and professional phagocytes), at five times post-infection.

Our data demonstrate that the large majority of the *S. aureus* isolates are facultative intracellular, with 187 of the 191 isolates being efficiently internalized in the four tested cell types. Moreover, visualization of the obtained datasets through clustering and dimensionality reduction using t-SNE captured a great diversity of intracellular fates of the bacterial isolates following internalization, most notably: (i) low replication, high persistence, and reduced induction of host cell death (group III); (ii) low/intermediate replication, with low persistence and high host cell death (groups IVa, IVc, and IVd); (iii) high replication, eliciting fast host cell death (group IVb); and (iv) high and prolonged replication associated with delayed host cell death (group V).

The phenotypic profiling analysis was combined with genome sequence analysis of the 191 isolates. Although genomic analysis is undoubtedly useful in pinpointing intrinsic pathogen features[52–54], phenotypical analysis, such as that performed in this study, provides critical information for the understanding of pathogen–host interactions. This is further demonstrated by the observation that BJI and Ba/IE isolates were segregated based on the multiparametric phenotypic profiling, but not on phylogenomic analysis. This is likely driven by the differences in replication (e.g., lower in BJI than in Ba/IE isolates in macrophages and epithelial cells), persistence (e.g., lower in BJI than in Ba/IE isolates in osteoblasts), and host cell viability (e.g., lower in BJI than in Ba/IE isolates in osteoblasts, higher in BJI than Ba/IE isolates in epithelial cells). However, the phenotypical analysis did not discriminate between IE and Ba isolates, in agreement with previous studies using a less complex set of phenotypes[20,55,56]. These results suggest that *S. aureus* from unrelated clinical sources present a set of distinctive phenotypic characteristics (BJI vs. Ba/IE), unlike those from more closely related clinical origins (IE is a complication occurring from Ba), at least based on the parameters and infection settings tested.

The phenotypic analysis highlighted the multifaceted modes of interaction of *S. aureus* with host cells, altogether demonstrating that drawing conclusions from the study of a few prototypical *S. aureus* strains can be misleading. Another layer of complexity is introduced by the distinct interplay of *S. aureus* isolates with different host cell types. Interestingly, the intracellular phenotypes described above are generally conserved in the infection of non-professional phagocytic cells, but highly divergent in macrophages. For example, in agreement with

previous reports[36,57–59], we observed that the *S. aureus* isolates present high persistence at 48 hpi upon infection of endothelial cells and macrophages. However, the pairwise comparison of the persistence of individual isolates in the two cell types shows a remarkable dissimilarity. This suggests that the host cell is key to defining the extent of persistence for each individual bacterial isolate.

In line with the multiple *S. aureus* fates within host cells, it is described that in non-professional phagocytes, some *S. aureus* isolates can escape from the vacuole to the host cytosol, while others are retained in the vacuole[36,37]. Analysis of the subcellular localization of the *S. aureus* isolates in epithelial cells showed that the majority of the isolates were indeed able to escape from the vacuole. As previously suggested[34,36,42], our data show that, in general, bacterial cytosolic localization correlates with high replication and rapid induction of host cell death. However, we have identified a group of atypical isolates that reach the cytosol but present delayed cell death (see below). Additionally, not all isolates that reach the cytoplasm show high replication, indicating that host cell death elicited by *S. aureus* can also occur independently of bacterial replication.

Among the various intracellular phenotypes identified, we focused our attention on two distinctive groups of isolates for downstream mechanistic studies. One group of isolates was phenotypically characterized by low vacuolar escape and replication, high intracellular persistence, and impaired induction of host cell death in non-professional phagocytes. All these isolates were shown to be *agr*-defective. However, no particular genetic features were identified in the *agr* locus of these isolates, suggesting the existence of alternative/multiple traits that converge in *agr*-deficiency. Indeed, *agr* deficiency can originate from mutations in the *agr* locus itself or in a complex network of upstream genes influencing *agr* expression[60]. Previous studies have shown that *agr*-defective isolates are detected in several types of infections, including osteomyelitis, bacteremia, and infective endocarditis, and are implicated in chronic and relapsing infections[61,62].

Of particular interest was the second group of isolates, which showed very high and prolonged intracellular replication associated with delayed onset of host cell death in non-professional phagocytic cells. Our data suggest that these isolates have an extended timeframe to more effectively colonize the host cell/organism, reaching higher levels of intracellular replication than other isolates with high replication, while leading to slower host cell/organism death. Presumably, these *S. aureus* isolates have evolved a unique mechanism to preserve their replicative niche, by delaying host cell death and thus allowing higher levels of bacterial intracellular replication. Mechanistically, we determined that this group of isolates are deficient in the cysteine protease staphopain A. Supporting our data, previous reports showed

that staphopain A induces host cell death in epithelial cells through a still unknown mechanism[44], and that mutants in its transcriptional regulator Rsp present prolonged residence within host cells, reduced cytotoxicity and lethality in a pneumonia or sepsis murine model[63]. Our study highlights the relevance of staphopain A to *S. aureus* pathogenicity, by demonstrating the existence of naturally occurring *scpA* loss-of-function mutants among the isolates collected from BJI, IE, and Ba patients. In our study, the prevalence of these isolates can be estimated at ca. 6–8%. The presence of staphopain A defective mutants in *S. aureus* isolates from other collections and clinical origins and the clinical outcome of infections by these mutants deserves further investigation.

Overall, the phenotypic and genomic datasets from this comprehensive large-scale study constitute a valuable and unique resource to further delve into the complexity of *S. aureus*−host crosstalk. Moreover, this systematic understanding of *S. aureus* intracellularity has relevant implications for the management and treatment of staphylococcal infections, since many of the most commonly used antibiotics are ineffective against intracellular pathogens[64,65]. As such, even in cases of bacterial culture tests that show a broad antibiotic susceptibility, the lack of intracellular effect of the drugs might lead to treatment failure, relapse, or chronic infections. Our study supports a change in the treatment paradigm, which should consider not only the antibiotic susceptibility profile but also the diverse intracellular lifestyles of *S. aureus* when deciding the course of therapy to effectively eliminate the pathogen.

## Methods

### Mammalian cell culture

Human epithelial HeLa-229 (ATCC, CCL-2.1), human osteosarcoma U2OS (ATCC, HTB-96), and HEK293T (ATCC, CRL-3216) cells were cultured in DMEM GlutaMax containing 1.0 g/l glucose (HyClone, SH30021.01), human endothelial EA.hy926 (ATCC, CRL-2922) cells were cultured in DMEM GlutaMax containing 4.5 g/l glucose (HyClone, SH30243.01), human monocyte THP1 (ATCC, TIB-202) cells were cultured in RPMI 1640 GlutaMAX (HyClone, SH30027.01). Media were supplemented with 10% fetal bovine serum (Gibco, 10270-106). Cell lines were acquired from ATCC/LGC Standards, and no further authentication was performed. Human umbilical vein endothelial cells (HUVEC; Lonza, C2519A) were maintained in EGM-2 Endothelial Cell Growth Medium-2 BulletKit (Lonza, CC-3162), according to the vendor's instructions. All cells were maintained at 37 °C in a humidified atmosphere with 5% $CO_2$. All cell lines tested negative for mycoplasma contamination.

For the generation of HeLa-229 cells stably expressing mCherry-Galectin-3, a lentiviral transfer plasmid was assembled in the pLJM1-EGFP backbone[66] (addgene #19319, gift from D. Sabatini). Briefly, mCherry-Galectin-3 was excised with AgeI and EcoRI from pmCherry-Gal3 (addgene #85662, gift from H. Meyer) and subcloned in the pLJM1 plasmid between the AgeI and EcoRI restriction sites. The HeLa-229 cells stably expressing mRFP-CWT escape marker were generated using the pLVTHM-H2B-BFP-IRES-mRFP-CWT plasmid (gift from M. Fraunholz).

For each construct, lentiviral particles were produced in HEK293T using psPAX2 and pMD2.G packaging plasmids (addgene #12260 and #12259, gift from D. Trono), using Lipofectamine 3000 (Life Technologies, L-3000015) according to the manufacturer's instructions. Viral supernatants were harvested at 48 and 72 h after transfection, combined and centrifuged at 350×*g*, filtered through a 0.45 µm PVDF filter, and used for transduction. Lentiviral transduction of HeLa was performed using various dilutions of the viral supernatants in 6-well plates (Corning, 3516) containing $1.5 \times 10^5$ cells per well in media supplemented with 8 µg/ml polybrene (Sigma, H9268). Puromycin selection (400 ng/ml) of HeLa cells transduced with pLJM-mCherry-Galectin-3 plasmid was started at 24 h after transduction and

maintained for an additional 72 h. No antibiotic selection was performed on HeLa cells transduced with pLVTHM-H2B-BFP-IRES-mRFP-CWT plasmid. The surviving cell populations were sorted in a FACSAria III cell sorter (BD Biosciences) to enrich for homogenous populations of mCherry-positive cells or BFP- and mRFP-positive cells. Cells transduced at low MOIs were used, to minimize multiple integration events.

### Bacterial isolates and strains

The *S. aureus* isolates collection used in this study was designed to include isolates collected from patients with a diversity of diseases in which *S. aureus* represents a major and relevant pathogen, namely: (i) bone/joint infections (BJI); (ii) bacteremia without infective endocarditis (Ba; based on negative requested trans-thoracic or trans-esophageal echocardiography, and not meeting post-hospital criteria for infective endocarditis at a 3-month follow-up visit); and (iii) infective endocarditis (IE). The 93 BJI isolates, all methicillin-susceptible, were responsible for a first episode of BJI at the Hospices Civils de Lyon, Lyon, France, from 2001 to 2011[21]; the 48 Ba and the 50 IE isolates belonged to the French national prospective multicenter cohort VIRSTA[22]. The *S. aureus* isolates are available from F.L. or F.V. upon reasonable request.

The colony forming unit assays, mammalian cell viability assays using ATPlite, and infections of HUVEC were performed with a subset of isolates from the *S. aureus* collection, specifically: cluster I – BJI006, BJI010, IE070, IE074; cluster II – BJI089, BJI111, Ba003, Ba033, IE051; cluster III – BJI035, BJI067, Ba039, Ba048, IE099; cluster IVa – BJI001, BJI009, BJI072, IE053, IE093; cluster IVb – BJI062, BJI079, IE059, IE060, IE095; cluster IVc – BJI051, BJI073, Ba044, IE083, IE091; cluster IVd – BJI060, BJI103, Ba031, IE061, IE090; cluster V – BJI008, BJI012, Ba009, IE055, IE057.

*S. aureus* USA300 SF8300 is a clinical strain of community-associated MRSA genotype USA300-0114[67]. *S. aureus* 8325-4 strain is a derivative of NCTC8325 cured of three prophages[68]. *S. aureus* DU5883 is an isogenic double *fnbA fnbB* mutant of *S. aureus* 8325-4[69]. *S. aureus* RN4220 is a nitrosoguanidine-induced mutant of 8325-4, phenotypically *agr*-negative[70,71]. *S. aureus* USA300 JE2 is a derivative of USA300 LAC, a community-associated MRSA, which was cured of three plasmids and used to generate the Nebraska Transposon mutant library[72]. The *S. aureus* insertional transposon mutants of *scpA* (NE1278; SAUSA300_1890) and of *agrA* (NE1532; SAUSA300_1992) were obtained from the Nebraska Transposon mutant library.

For expression of staphopain A in the *S. aureus* isolates deficient on staphopain A expression, the *scpAB* operon including the native promoter region (361 bp upstream of the start codon) and the transcription termination signal (269 bp downstream of the stop codon) was amplified by PCR from *S. aureus* genomic DNA (*S. aureus* USA300) using the following primer pair 5′-AAACTGCAGTATTC-TATTGCATAGGTGTGGCATT-3′ and 5′-CGGGATCCCTATTTGAA-GAGGAAAGGCTATTC-3′, as previously described[44]. The product was cloned in the pJL74 plasmid using the PstI and BamHI restriction sites. The resulting plasmid was verified by DNA sequencing and passaged through *E. coli* IM08B and IM01B plasmid artificial modification strains (gift from I. Monk). These strains mimic the type I adenine methylation profiles of *S. aureus* CC8 and CC1, respectively, in order to bypass the *S. aureus* restriction-modification barrier[73]. The plasmid purified from these *E. coli* strains was transformed into the 13 *S. aureus* clinical isolates belonging to group V and *scpA* mutant, by electroporation[74]. Transformants were checked by colony PCR and verified by DNA sequencing.

For the expression of GFP in selected *S. aureus* strains/clinical isolates used for the time-lapse microscopy, a pJL74-derivative plasmid harboring a tetracycline resistance gene (tet) and a *gfp* gene was engineered. A single DNA fragment containing the *tet* cassette and the *gfp* gene was cloned in the pJL74 backbone using the ApaI and XhoI restriction sites, and propagated in the IM08B *E. coli* strain. The

plasmid was purified and transformed into the clinical isolates by electroporation[74]. Transformants were checked by colony PCR and verified by the production of GFP using a Blue light LED transilluminator.

S. aureus was grown aerobically in Trypticase Soy broth (TSB; BD Biosciences, 211771) at 37 °C. When appropriate, the medium was supplemented with erythromycin 5 μg/ml or tetracycline 5 μg/ml.

## Fluorescence microscopy-based S. aureus infection assays

For time-course infection experiments, cells were seeded in black, clear-bottom 384-well plates (Greiner, 781090). HeLa and THP1 cells were plated 72 h before infection, at a density of $1.2 \times 10^3$ and $6 \times 10^3$ cells per well, respectively. THP1 cells were differentiated into macrophage-like cells by treatment with 50 ng/ml phorbol 12-myristate 13-acetate (Sigma, 79346), at the time of plating. HUVEC, U2OS, and EA.hy926 cells were plated 48 h before infection at a density of $1.25 \times 10^3$ (HUVEC) or $1.6 \times 10^3$ (U2OS, EA.hy926) cells per well.

For S. aureus infections, bacterial clinical isolates were grown overnight in round-bottom 96-well plates (Corning, 3799) at 37 °C with shaking. Overnight bacterial growths were diluted 1:100 in TSB and grown for 2 h 15 min at 37 °C with shaking, allowing the majority of bacterial cultures to reach $OD_{600}$ 0.3–0.8 (exponential phase). A duplicate of each bacterial isolate was grown for an additional 15 to 30 min intervals to ensure that all bacterial isolates were within the $OD_{600}$ 0.3–0.8 range. $OD_{600}$ was determined using an EnSpire Multimode Plate Reader (PerkinElmer). From the two growths of each bacterial isolate, infection with the growth showing $OD_{600}$ closer to 0.4 was chosen for the analysis. Infections were performed at a multiplicity of infection (MOI) ~25 by adding 10 μl of bacterial suspensions previously diluted in complete medium to the mammalian cells. Cells were centrifuged at $500 \times g$, at room temperature for 10 min, and incubated at 37 °C in a 5% $CO_2$ humidified atmosphere for 50 min. Extracellular bacteria were killed by replacing the medium with fresh medium containing 5 μg/ml lysostaphin (Ambi Products LLC, LSPN-50) and 100 μg/ml gentamicin (Sigma, G1272); all S. aureus isolates used are susceptible to gentamycin. Medium supplemented with antibiotics was maintained until the indicated times of collection. Two prototypical S. aureus strains, specifically USA300 and 8325-4, were used for standardization of the assay and included in all assay plates. To minimize systematic effects within plates (e.g., edge effects), the wells of the two outermost rows and the outermost columns of 384-well plates were left untreated, and only internal wells were used for infection/ analysis.

For S. aureus phagosomal escape and galectin-3 recruitment assessment, HeLa cells stably expressing mRFP-CWT or mCherry-Galectin-3 (prepared as described above) were seeded 48 h before infection in black, clear-bottom 384-well plates at a density of $1.5 \times 10^3$ cells per well. Dilution of bacterial cultures and infections were performed as described above.

For S. aureus isolates ectopically expressing staphopain A and E-64d treatment experiments, $1.5 \times 10^3$ cells (HeLa, U2OS, EA.hy926) per well were seeded in black, clear-bottom 384-well plates, 48 h prior to infection. When indicated, HeLa cells were treated with 10 μM E-64d (Santa Cruz Biotechnology, sc-201280) 1 h prior to infection and during infection. S. aureus overnight cultures were diluted 1:100 in TSB and grown at 37 °C with shaking until $OD_{600}$ 0.4. Bacteria were then harvested by centrifugation for 2 min at $12,000 \times g$ and resuspended in complete medium. Infections were performed as described above.

For the eFluor proliferation assay, $1.5 \times 10^3$ HeLa cells per well were seeded in black, clear-bottom 384-well plates, 48 h prior to infection. Labeling of S. aureus with eFluor-670 cell proliferation dye, an amine-reactive dye, was performed as previously described[25,26]. Briefly, bacteria were grown as described above, followed by labeling with 0.5 μg/ml eFluor-670 dye (eBiosciences, 65-0840-90) in PBS for 5 min at RT. Bacteria were then harvested by centrifugation at $12,000 \times g$ for

2 min, and resuspended in LB media for 3 min to quench unreacted dye. Bacteria were washed twice with PBS and then resuspended in complete medium. Infections with the labeled bacteria were performed as described above.

For time-lapse microscopy, infections were performed as described above with GFP-expressing S. aureus. Nuclei counterstaining with Hoechst 33342 (1:50,000, Thermo Fisher Scientific, H3570) was performed during bacterial incubation with host cells, prior to antibiotic addition. Extracellular bacteria were killed by replacing the medium with imaging medium (1:1 mix of DMEM (Gibco, A14430-01) with FluoroBrite DMEM (Gibco, A18967-01), supplemented with sodium pyruvate 1 mM (Gibco, 11360-070), 1x GlutaMax (Gibco, 35050-038) and 25 mM HEPES (pH 7.0)), containing 5 μg/ml lysostaphin (Ambi Products LLC, LSPN-50) and 100 μg/ml gentamicin (Sigma, G1272).

## Staining and immunofluorescence

At the indicated times post-infection, cells seeded in 384-well plates were rinsed with PBS and subsequently fixed with 4% paraformaldehyde for 15 min at room temperature or overnight at 4 °C (HeLa -mRFP-CWT). After 30 min of permeabilization with 0.5% Triton X-100 in PBS, S. aureus was labeled for 2 h at room temperature with 0.25 μg/ml BODIPY-FL vancomycin (Invitrogen, V34850), a glycopeptide antibiotic (conjugated to a fluorescent dye) that specifically binds to the cell wall of Gram-positive bacteria and efficiently labels intracellular S. aureus, upon host cell membrane permeabilization[16,17]. Cells were washed, and nuclei were counterstained with Hoechst 33342 (1:5,000, Thermo Fisher Scientific, H3570) for 15 min at room temperature. For staining of acidic vesicles, cells were treated for 30 min at 37 °C prior to fixation with 75 nM LysoTracker Red DND-99 (Life Technologies, L-7528).

## Image acquisition and analysis

Image acquisition was performed using an Operetta automated high-content screening fluorescence microscope (PerkinElmer), at 20× magnification, with a total of nine images acquired per well (number of mock-treated cells analyzed per experimental condition, independent experiment, and cell type are indicated in Supplementary Data 1).

Image analysis to quantify infection, bacterial replication, host cell viability, S. aureus vacuolar escape (S. aureus/CWT colocalization, infected cells positive for galectin-3 foci, and S. aureus/LysoTracker colocalization) were performed using custom workflows implemented in Columbus image analysis software (PerkinElmer), as previously described[19].

To quantify S. aureus infection and replication, segmentation of the nucleus and cytoplasm and identification of the regions with bacteria fluorescence signal within cells was performed using the building blocks "Find Nuclei", "Find Cytoplasm", and "Find Spots" implemented in Columbus software. The image segmentation procedure is based on a combination of global and individual thresholds, and other parameters (e.g., split factor, area, and contrast). Of note, nuclei (primary object) were first segmented based on Hoechst staining; host cell cytoplasm was also segmented in this channel, taking advantage of the faint cytoplasmatic background staining of Hoechst. Cells touching the edge of the images were excluded from further analysis. After the segmentation steps, intensity and morphological features (including integrated intensity and area) from the vancomycin BODIPY channel (green) were extracted using the 'Calculate Intensity Properties' and "Calculate Morphology Properties". In order to classify cells as infected/non-infected or with high bacterial replication (cells with high intracellular S. aureus load), we applied the "Select Population" building block with a set of cutoff criteria based on the extracted features (area and integrated intensity). Specifically, for each cell line and independent experiment, the cutoff criteria were adjusted based on the iterative analysis and visual inspection of a number of images from cells infected with the reference strain (USA300) at 1.5 hpi, and

subsequently applied to all images from all time points. This approach and the correct setting of cutoff criteria were validated and concordant with results obtained using alternative experimental approaches (CFU assays, eFluor).

For the evaluation of *S. aureus* vacuolar escape using the CWT reporter, after the selection of the infected cells as described above, mRFP-CWT spots were identified within the cytoplasm of infected cells using the "Find spots" building block. The area of mRFP/CWT spots was then restricted to the area of *S. aureus* determined as described above, and these areas were used to calculate mRFP-CWT/*S. aureus* colocalization ratios.

The quantification of colocalization ratios of *S. aureus* with Lyso-Tracker was achieved by detecting all LysoTracker-stained vesicles within the host cytosol and by using the area of bacterial and Lyso-Tracker vesicles, essentially as described above for *S. aureus*/CWT colocalization analysis.

To assess the percentage of infected cells presenting galectin-3 foci, following the selection of *S. aureus* infected cells as described above, galectin-3 foci were identified from the diffusely distributed fluorescence signal using the "Find spots" building block and enumerated. Cells were classified as positive or negative for galectin-3 recruitment based on the number of galectin-3 spots per cell; cells presenting more than 1 galectin-3 foci per cell were considered positive.

In all experiments involving the quantification of mRFP-CWT, LysoTacker, and galectin-3, the escape-proficient *S. aureus* USA300 and the escape-deficient *agrA* mutant were used as positive and negative controls, respectively. Cells with small nucleus size and high-intensity levels of Hoechst and mCherry signals, indicative of cell death, were excluded from the analysis.

For time-lapse microscopy, image acquisition was performed using an Operetta automated high-content screening fluorescence microscope (PerkinElmer), at 20× magnification, with a total of 4 images being acquired per well during 24 h with intervals of 30 min.

### Colony forming unit (CFU) assays
HeLa cells were seeded in 24-well plates (Corning, 3536) 48 h before infection at a density of $5 \times 10^4$ cells per well. *S. aureus* overnight cultures were diluted 1:100 in TSB and grown at 37 °C with shaking until $OD_{600}$ 0.4. Bacteria were then harvested by centrifugation for 2 min at $12,000 \times g$ and resuspended in complete medium of the mammalian cells. Infections were performed at an MOI of 25. Following the addition of the bacteria, cells were centrifuged at $500 \times g$, at RT for 10 min, and incubated at 37 °C in a 5% $CO_2$ humidified atmosphere for 50 min. Extracellular bacteria were killed by replacing the medium with fresh medium containing 5 μg/ml lysostaphin (Ambi Products LLC, LSPN-50) and 100 μg/ml gentamicin (Sigma, G1272). At the indicated times postinfection, cells were washed with PBS and lysed with PBS containing 0.1% Triton X-100 (Roth, 3051.4). Cell lysates were serially diluted in PBS and plated on TSB agar plates.

### Mammalian cell viability assay
HeLa cells were seeded in black clear-bottom 384-well plates 48 h before infection at a density of $1.5 \times 10^3$ cells per well. Bacterial growth and infections were performed as described above in "Fluorescence microscopy-based S. aureus infection assays". Host cell viability was assessed using the ATPLite 1step Luminescence Assay System (PerkinElmer, 6016731), according to the manufacturer's instructions. Briefly, at the indicated times post-infection, cells were rinsed with PBS, and 25 μl of ATPlite reagent was added per well. Luminescence was measured using an EnSpire Multimode Plate Reader (PerkinElmer).

### RNA isolation and quantitative real-time PCR
For total RNA isolation, overnight *S. aureus* cultures were diluted 1:100 in TSB and grown at 37 °C with shaking for 8 h (stationary phase).

Bacterial cultures were centrifuged for 5 min at $12,000 \times g$, bacterial pellets were lysed in TRIzol (Invitrogen, 15596026) and RNA was extracted by phenol-chloroform followed by isopropanol precipitation. For quantification of gene expression, 0.5 μg of total RNA was reverse transcribed using hexameric random primers and M-MLV reverse transcriptase (Invitrogen, 28025021), according to the manufacturer's instructions. qRT-PCR was performed using SsoAdvanced Universal SYBR Green Supermix (Bio-Rad, 172-5274) according to the manufacturer's instructions, using a CFX96 TouchTM Real-Time PCR detection system (Bio-Rad). The following primer pairs were used: RNAIII 5′-GAAGGAGTGTTTCAATGG-3′ and 5′-TAAGAAAATACATAGC ACTGAG-3′[75]; psm-α 5′-GGCCATTCACATGGAATTCGT-3′ and 5′-GCC ATCGTTTTGTCCTCCTG3′[33]; GAPDH 5′-TACACAAGACGCACCTCACA GA-3′ and 5′-ACCTGTTGAGTTAGGGATGATGTTT-3′[76]. Expression was normalized to GAPDH, and the relative gene expression was calculated using the $2^{-\Delta\Delta Ct}$ method.

### Protein extracts and western blot
To prepare *S. aureus* supernatant samples for western blot, overnight bacterial cultures were diluted 1:100 in TSB and grown at 37 °C with shaking for 4 h. Bacterial cultures were then adjusted to $OD_{600}$ 4 and 20 ml of the cultures were centrifuged for 10 min at $4200 \times g$ at 4 °C. The bacterial supernatants were filtered through 0.22 μm syringe filters and the bacterial secreted proteins were precipitated overnight at −20 °C in 10% trichloroacetic acid (Roth, 8789.1). Proteins were collected by centrifugation ($4200 \times g$, 45 min, 4 °C) and washed twice with ice-cold 100% acetone followed by centrifugation ($15,000 \times g$, 15 min, 4 °C). Dried protein pellets were resuspended in sample buffer (50 mM HEPES (pH 7.0), 250 mM NaCl, 1 mM DTT, 0.01% Triton X-100) and protein concentration was determined with NanoDrop 2000 spectrophotometer (Thermo Fisher). Protein lysates (~200 μg) were diluted in Laemmli sample buffer, incubated for 10 min at 95 °C, and separated in SDS-PAGE followed by western blotting. The anti-staphopain A antibody (1:1000; antibodies-online, ABIN967004; RRID:AB_2894409) and an anti-rabbit secondary antibody coupled to horseradish peroxidase (1:10,000, GE Healthcare, NA934) were used. Signal was detected using enhanced chemiluminescence (ECL), using an Imager Chemi 5QE CCD camera (VWR).

### *Galleria mellonella* infection
Final instar larval stage *G. mellonella* were purchased from UK Waxworms and stored in the dark at 16 °C. Only active larvae with no visible signs of melanization and weighing 250–300 mg were used in the experiments. Groups of ten larvae were randomly assigned and placed in 10-cm Petri dishes and incubated at 37 °C for 24 h before bacterial inoculation. Individual larvae were infected with $2 \times 10^6$ CFU of *S. aureus* by intrahemocelic injection of 10 μL of inoculum, in the last left proleg using a Hamilton precision syringe with a 30-gauge 12-mm long needle. *S. aureus* inoculum was prepared as described above and diluted in PBS. After injection, infected larvae were incubated at 37 °C and the survival of each group was scored daily for 7 days. Larvae were considered dead when they displayed melanization and no movement in response to touch. Four independent experiments were performed with 10 larvae per clinical isolate. As control groups, larvae were inoculated with vehicle (PBS) or non-injected, with a maximum of one dead larva observed in these groups by day 7; for simplicity, results for these groups were not included in Fig. 4h.

### Fibronectin adhesion assays
The fibronectin adhesion assay was performed in 96-well flat-bottom plates as previously described[77], with minor changes. Briefly, the wells were coated with 200 μl of human fibronectin (Corning, 356008) at 50 μg/ml (18 h, 4 °C). Wells were then washed three times with PBS supplemented with 1% FBS (20 min, 37 °C). Overnight *S. aureus* cultures were adjusted to $OD_{620}$ 1.0, corresponding

to ca. $1 \times 10^9$ CFUs/ml. One ml of each bacterial suspension was centrifuged (12,000 × g, 2 min), and pellets were washed and resuspended in PBS. 100 µl of each bacterial suspension were added to the wells of the fibronectin-coated plate and incubated for 30 min at 37 °C with mild shaking. Wells were washed three times with PBS to remove non-adherent bacteria. Adherent bacteria were fixed with glutaraldehyde (2.5% v/v in 0.1 M phosphate buffer for 2 h at 4 °C) and stained with crystal violet (0.1% m/v; Elitech, SS-041C-EU) for 30 min at room temperature. After three washes with PBS, the total remaining stain impregnating the adherent bacteria was solubilized using 100 µl of 0.2% Triton X-100 at room temperature for 30 min. Quantification of adherent bacteria was performed by measuring $OD_{590}$ using an Infinite M Nano+ microplate reader (Tecan). The values were normalized to the reference strain *S. aureus* 8325-4; *S. aureus* DU5883 was used as a control.

### Whole-genome sequencing and phylogenetic analysis

*S. aureus* isolates were subcultured twice from frozen glycerol broth onto horse blood agar. Extraction of genomic DNA from single colonies was preceded by treatment with lysozyme (20 mg/ml; EUROMEDEX, 5933-C) and lysostaphin (100 µg/ml; Sigma, L7386) for 30 min at 37 °C. DNA extraction was performed using the Maxwell RSC Blood DNA Kit (Promega, AS1400), DNeasy UltraClean 96 Microbial Kit (QIAGEN, 10196-4), or DNeasy Blood & Tissue Kit (QIAGEN, 69504), according to the manufacturer´s instructions. Nextera XT DNA Library Preparation Kit (Illumina, FC-131-1096), Illumina DNA Prep (Illumina, 20018705), or TruSeq DNA PCR-Free (Illumina, 20015963) were used for the library preparation. Whole-genome sequencing was conducted on MiSeq, HiSeq 1500, HiSeq 4000, or NextSeq 500 platforms (Illumina) with a read length of 2 × 250 bp for MiSeq or 2 × 150 bp for the other platforms.

The reads were trimmed using Cutadapt (v3.4) and Trimmomatic (v0.39)[78] in paired-end mode. Genome de novo assembly was performed using SPAdes (v3.13.0)[79], with options "--careful" and "--cov-cutoff 10". Assembly quality was evaluated using QUAST (v5.0.2)[80]. Genomes were annotated with Prokka (v1.14.5)[81] with default parameters. The mean sequencing depth calculated with SAMtools (v1.15)[82] was 566× (based on the sampling of 95 genomes, on the three longest scaffolds for each genome). The .gff files generated by Prokka were used in Roary (v3.13.0)[83] to perform a pan-genome analysis. This analysis was conducted with parameters "-e --mafft". The genes from the core genome (1699 genes), obtained with Roary, were used to build the phylogenetic tree with FastTree (v2.1.10)[84] with default settings. FastTree infers approximately maximum-likelihood phylogenetic trees from nucleotide sequences.

Mutations in *fnbA* and *scpA* genes in isolates from clusters I and V, respectively, were identified by performing multiple sequence alignments with MAFFT(v7)[85] and comparison to the corresponding sequences of the *S. aureus* reference strain 8325 (NC_007795.1). Mutations were manually identified by the analysis of the alignments.

Agr type determination was performed by comparison of the AgrB, AgrC, and AgrD sequences of the 191 *S. aureus* clinical isolates genomes to an *agr* sequence database containing the sequences of the four *agr* types, specifically COL (*agr* type I; CP000046.1), Mu50 (*agr* type II; BA000017.4), MRSA252 (*agr* type III; BX571856.1), and RN5881 (*agr* type IV; AF288215.1). Blastn (v2.12.0+)[86] was used to perform the sequence alignments. Agr type assignment was manually performed according to the best blast hits.

The *S. aureus* clinical isolates genome sequencing data have been deposited in the European Nucleotide Archive (ENA), accession number PRJEB48298.

### Statistical/data analysis

Unless otherwise indicated, data are presented as mean or mean ± standard error of the mean (s.e.m.), with the exact number of experiments performed indicated in figure legends. Microsoft Excel (Microsoft Office 2013) was used to compile the data, and statistical analysis was performed using Prism software (v7.00; GraphPad). The normal distribution of the data were assessed by the Shapiro–Wilk test. The following tests were used for statistical comparison of datasets: from two conditions/groups, two-tailed Student's *t*-test (for parametric data) or Wilcoxon signed-rank test (non-parametric data); from three or more groups, one-way ANOVA with Tukey's post hoc test or Dunnett's post hoc test (for parametric data), or Kruskal–Wallis with Dunn's multiple comparison test (non-parametric data); for the survival curves, log-rank (Mantel–Cox) test. Values of $P < 0.05$ were considered significant. Statistical analyzes are detailed in Supplementary Data 3.

Heat maps of the phenotypic profiles of individual *S. aureus* isolates and hierarchical clustering (method UPGMA, Euclidean distance) were performed using TIBCO Spotfire Analyst software (v.10.10.3; TIBCO). t-SNE analysis was performed using the Interactive t-SNE web tool (https://jefworks.github.io/tsne-online/ developed by J. Fan, Johns Hopkins University), with perplexity 25, learning rate 20, max iterations 1000, and Euclidean distance. Prior to t-SNE analysis all features were normalized to be in the range of 0–100. Sankey plot was created using the SankeyMATIC web tool (https://sankeymatic.com/ developed by Steve Bogart). Adobe Photoshop (Adobe, CS6, CC) and Adobe Illustrator (Adobe, CS6, CC) were used for assembling microscopy images and figures, respectively.

### Reporting summary

Further information on research design is available in the Nature Portfolio Reporting Summary linked to this article.

## Data availability

The data corresponding to the *S. aureus* phenotypic screenings are provided in the supplementary information (Supplementary Data 1, 2); raw microscopy datasets and custom image analysis workflows implemented in Columbus image analysis software are available from the corresponding authors upon reasonable request; the *S. aureus* clinical isolates genome sequencing data have been deposited in the European Nucleotide Archive (ENA), accession number PRJEB48298, available at https://www.ebi.ac.uk/ena/browser/view/PRJEB48298. The genome sequences or *agr* locus sequence of *S. aureus* strains used in this study are available in the National Center for Biotechnology Information (NCBI) GenBank under accession numbers NC_007795.1 (*S. aureus* 8325; https://www.ncbi.nlm.nih.gov/nuccore/NC_007795), CP000046.1 (*S. aureus* COL; https://www.ncbi.nlm.nih.gov/nuccore/CP000046), BA000017.4 (*S. aureus* Mu50; https://www.ncbi.nlm.nih.gov/nuccore/BA000017.4), BX571856.1 (*S. aureus* MRSA252; https://www.ncbi.nlm.nih.gov/nuccore/BX571856.1), and AF288215.1 (*S. aureus* RN5881; https://www.ncbi.nlm.nih.gov/nuccore/AF288215.1). Source data are provided with this paper.

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

## Acknowledgements

I.R.L. and R.J.S. are recipients of PhD fellowships (PD/BD/146464/2019 and PD/BD/129294/2017) of the Doctoral Program in Experimental Biology and Biomedicine of the Center for Neuroscience and Cell Biology, University of Coimbra. We thank Martin J. Fraunholz (University of Würzburg) for providing the pLVTHM-H2B-BFP-IRES-mRFP-CWT

plasmid, and Ian Monk (University of Melbourne, Australia) for the *E. coli* IM08B and IM01B strains. This work was supported by grants from the European Union's Horizon 2020 research and innovation program (under the Marie Skłodowska-Curie grant agreement No 893942 to L.M.A), ERA-NET Infect-ERA StaphIN (031L0094, BMBF, Germany, to A.E.; Infect-ERA/0001/2015, FCT, Portugal, to M.M.; PCIN-2015-151, MINECO, Spain, to D.L.; and ANR 15-IFEC-0002-04, France, to F.L. and F.V), and the ERDF—European Regional Development Fund through COMPETE 2020 and Portuguese Foundation for Science and Technology (POCI-01-0145-FEDER-007440, UIDB/04539/2020, POCI-01-0145-FEDER-029999 to M.M. and A.E.).

## Author contributions

I.R.L. performed and analyzed the data from the *S. aureus* clinical isolates infection screenings and phenotypic characterization, and participated in manuscript writing; L.M.A. contributed to the *S. aureus* clinical isolates phenotypic characterization; I.R.L. and R.J.S. established the stable cell clones; E.P.V, A.M.C., and D.L. generated the *S. aureus* strains ectopically expressing staphopain A or GFP; J.J. and M.B. performed the whole genome and phylogenetic analysis; J.J. performed the fibronectin-binding assays; F.L. and F.V. provided the panel of *S. aureus* clinical isolates and supervised the WGS and phylogenetic analysis; M.M. and A.E. coordinated the work and wrote the manuscript, with input from all the authors.

## Competing interests

The authors declare no competing interests.

## Ethics

The 93 *S. aureus* BJI isolates were passively collected in routine at Institute for Infectious Agents—Hospices Civils de Lyon. As no personal data were collected in the context of the present study, the use of the sole clinical isolates for research purposes requires no specific agreement from patients, according to the French health authorities. The 48 Ba and 50 IE *S. aureus* isolates were collected from the French national prospective multicenter cohort VIRSTA (LeMoing, 2015) registered in the European Clinical Trials Database (EUDRACT 169 2,008-A00680-55).
