## [Peer Review File · Nature Communications]

Microscopy-based phenotypic profiling of infection by *Staphylococcus aureus* clinical isolates reveals intracellular lifestyle as a prevalent featureEditorial Note: This manuscript has been previously reviewed at another journal that is not operating a transparent peer review scheme. This document only contains reviewer comments and rebuttal letters for versions considered at Nature Communications.

Reviewer #1 (Remarks to the Author):

The authors have made huge strides in order to improve the quality and breadth of the original manuscript. While I still have issues with the overall novelty of the study, the authors are correct in that the present study is the first study that provides a holistic, system biology approach to study invasion and intracellular replication by *S. aureus*. The authors are to be applauded by taking seriously the main critiques by the prior reviewers (i.e. a lack of validation with other techniques and overstatements). The new data validating the main original findings provide further credence to the main messages of the study.

Overall, this is a tour de force study providing a unique resource to the community.

Reviewer #3 (Remarks to the Author):

The authors have done extensive work and have rewritten the paper to address the main comments of all reviewers. As I said before, I find that this manuscript provides an important and comprehensive resource for the bacterial pathogenesis field, cementing the relevance and prevalence of the intracellular lifestyle of *S. aureus* during infection. The current version of the manuscript is also significantly improved, fixing open/weak points of the first version. Kudos to all authors.

My only main point is that the authors can weave in more some of the answers they give to reviewers into the manuscript - e.g. some of replies to technical questions (gentamicin sensitivity of all strains; Vancomycin BODIPY) are answered to reviewers but never make it to the text.

As a more minor point. although I agree with the authors' explanation about ED64-d (and that in this case effect seems to be ScpA-dependent), I would like to point out that for other pathogens, cysteine protease inhibitors have been shown to decrease host cell death and cysteine proteases to affect course for infection (see for example PMID 32514074 for *Salmonella*). So in their new sentence, authors may want to bring this up, and be more conservative in their statement - "our results (strongly) suggest rather than demonstrate"

Microscopy-based phenotypic profiling of infection by *Staphylococcus aureus* clinical isolates reveals intracellular lifestyle as a prevalent feature

Reviewers Comments: black

Our reply: green

Reviewer #1 (Remarks to the Author):

The authors have made huge strides in order to improve the quality and breadth of the original manuscript. While I still have issues with the overall novelty of the study, the authors are correct in that the present study is the first study that provides a holistic, system biology approach to study invasion and intracellular replication by *S. aureus*. The authors are to be applauded by taking seriously the main critiques by the prior reviewers (i.e. a lack of validation with other techniques and overstatements). The new data validating the main original findings provide further credence to the main messages of the study.

Overall, this is a tour de force study providing a unique resource to the community.

We are very grateful to the Reviewer for acknowledging our efforts to improve the manuscript, and his/her positive evaluation of our work.

Reviewer #3 (Remarks to the Author):

The authors have done extensive work and have rewritten the paper to address the main comments of all reviewers. As I said before, I find that this manuscript provides an important and comprehensive resource for the bacterial pathogenesis field, cementing the relevance and prevalence of the intracellular lifestyle of *S. aureus* during infection. The current version of the manuscript is also significantly improved, fixing open/weak points of the first version. Kudos to all authors.

We are very grateful to the Reviewer for his/her positive evaluation of our work.

My only main point is that the authors can weave in more some of the answers they give to reviewers into the manuscript - e.g. some of replies to technical questions (gentamicin sensitivity of all strains; Vancomycin BODIPY) are answered to reviewers but never make it to the text.

As recommend by the Reviewer, we have introduced in the revised version of the manuscript the information regarding the gentamycin susceptibility for all isolates (line 515), and accessibility/labelling with Vancomycin BODIPY of intracellular *S. aureus* (line 552).

As a more minor point. although I agree with the authors' explanation about ED64-d (and that in this case effect seems to be ScpA-dependent), I would like to point out that for other pathogens, cysteine protease inhibitors have been shown to decrease host cell death and cysteine proteases to affect course for infection (see for example PMID 32514074 for Salmonella). So in their new sentence, authors may want to bring this up, and be more conservative in their statement - "our results (strongly) suggest rather than demonstrate"

We modified the sentence as suggested by the Reviewer, and introduced the information regarding the impact of host cysteine protease inhibition on bacterial infection, e.g. Salmonella (lines 326-329).